# Robust and interpretable prediction of gene markers and cell types from spatial transcriptomics data

Xiao Tan[1,2], Onkar Mulay[1,2], Jacky Xie[3], Samual MacDonald[4,5], Taehyun Kim[6], Chenhao Zhou [7], Zherui Xiong[1,2], Samuel X. Tan[7], Nan Ye[3], Amy McCart Reed [8], Kiarash Khosrotehrani [7], Fred Roosta[3,4], Maciej Trzaskowski[1,5,9], Peter T. Simpson [8,10] & Quan Nguyen [1,2] ✉

Spatial transcriptomics (ST) links tissue morphology with gene expression values, opening new avenues for digital pathology. Deep learning models are used to predict gene expression or classify cell types directly from images, offering significant clinical potential but still requiring improvements in interpretability and robustness. We present STimage as a comprehensive suite of models to predict spatial gene expression and classify cell types directly from standard H&E images. STimage enhances robustness by estimating gene expression distributions and quantifying both data-driven (aleatoric) and model-based (epistemic) uncertainty using an ensemble approach with foundation models. Interpretability is achieved through attribution analysis at single-cell resolution integrated with histopathological annotations, functional genes, and latent representations. We validated STimage across diverse datasets, demonstrating its performance across various platforms. STimage-predicted gene expression can stratify patient survival and predict drug response. By enabling molecular and cellular prediction from routine histology, STimage offers a powerful tool to advance digital pathology.

Haematoxylin and Eosin (H&E) staining of tissue samples has been used by pathologists for more than a century. However, microscopic examination of H&E images is often time-consuming and variable[1]. Moreover, a range of molecularly distinct cell types may not be distinguishable by eye[2]. Recent FDA approvals[3] of whole-slide imaging (WSI) for primary diagnostic use have encouraged more widespread adoption of digital WSI[4]. However, the development and optimisation of automated cancer classification models requires a large number of pathologist-annotated tissue images[5,6]. HE2RNA[7] was a pioneering model trained without pathological annotation. Instead, HE2RNA uses matched H&E images and RNA-seq data from the TCGA database for 8725 patients across 28 different cancer types, to predict gene expression at the tile level (each WSI H&E image is split into multiple tiles). However, due to the nature of the data input, HE2RNA lacks ground truth at the tile level, relying instead on the aggregated gene expression value from the whole image (i.e., all tiles in the image have the same ground truth for one gene), thus limiting its ability to assess tumour heterogeneity.

[1]Genomics and Machine Learning Lab, Institute for Molecular Bioscience, St Lucia, QLD, Australia. [2]Queensland Institute of Medical Research Berghofer; QIMRB National Centre for Spatial Tissue and AI Research (NCSTAR), Herston, QLD, Australia. [3]School of Mathematics and Physics, The University of Queensland, St Lucia, QLD, Australia. [4]ARC Training Centre for Information Resilience, The University of Queensland, St Lucia, QLD, Australia. [5]Max Kelsen, Spring Hill, QLD, Australia. [6]Pathology Queensland, Royal Brisbane & Women's Hospital, Herston, QLD, Australia. [7]Frazer Institute, Faculty of Health, Medicine and Behavioural Sciences, The University of Queensland, Woolloongabba, QLD, Australia. [8]UQ Centre for Clinical Research, Faculty of Health, Medicine and Behavioural Sciences, The University of Queensland, Herston, QLD, Australia. [9]IntelMagik, West End, QLD, Australia. [10]School of Biomedical Sciences, Faculty of Health, Medicine and Behavioural Sciences, The University of Queensland, St Lucia, QLD, Australia. ✉e-mail: quan.nguyen@qimrb.edu.au

Spatial transcriptomics (ST-seq) produces both imaging and sequencing information and has the unique capability of measuring over 20,000 genes without tissue dissociation, preserving tissue anatomy and the microenvironmental context[8]. However, spatial transcriptomics remains prohibitively expensive for clinical applications. To improve scalability, researchers have attempted to predict spatial gene expression using deep learning methods, including STnet[9], Hist2Gene[10], Hist2ST[11], and DeepSpace[12]. However, these methods are still limited in interpretability, robustness, and accuracy. Crucially, clinical applications require models to be understandable to clinicians and patients for safety, trust, validation, and therapeutic decision-making[13].

Here, we report STimage, a deep learning probabilistic framework with reduced uncertainty and improved robustness and interpretability, applicable even in cases where training datasets are limited. We tested STimage across three different cancer datasets, including breast, skin, and kidney cancer, as well as a liver immune disease dataset, assessing the performance of predicting the most variable genes or functional genes. The shared biological relationships between these genes are learned simultaneously, allowing us to assess the performance of gene panels. We introduced a geometric assessment metric that better represents the model's ability to predict spatial patterns.

## Results

### STimage model - distribution prediction, ensemble, interpretability and uncertainty estimates

First, in the STimage pipeline, we implement the image preprocessing step (Fig. 1a). STimage performs stain normalisation to account for technical variation between image inputs used during model training. To remove uninformative tiles, we calculate tissue coverage and filter out tiles with low coverage. In STimage, image tiles are either used to fine-tune the ResNet50 model (as a minibatch of 64 tiles by default) or passed through pretrained foundation models to be transformed into feature embeddings (Fig. 1b). All latent features are then connected to a layer of neurons representing the parameters of the negative binomial (NB) distribution for each gene, which is optimised using a log-likelihood loss function. Alongside the distribution estimates, we also quantify model prediction uncertainty (Fig. 1b). This is a critical feature not available in existing tools, yet essential given the high variability expected in cancer datasets. Multiple genes can be trained simultaneously, with each gene corresponding to a pair of NB parameters. By sampling from the NB distribution, STimage can learn to predict the expression of individual genes or gene groups. Importantly, we incorporate model interpretability via Local Interpretable Model-agnostic Explanations (LIME) (Fig. 1b).

Several existing programs for gene expression prediction, such as STnet[9], Hist2ST[11], Hist2Gene[10], DeepSpace[12], and Xfuse[14], produce fixed-point estimates without any uncertainty quantification. In contrast, STimage introduces an approach to quantify and mitigate uncertainties in gene expression predictions, distinguishing between uncertainty due to model knowledge gaps (epistemic) and inherent variability in the data (aleatoric) (Fig. 1b).

### Prediction performance metrics and STimage accuracy

To assess model performance, we first visually compared the prediction of an abundant cancer marker (*COX6C*), broadly expressed across the tissue, with a less abundant marker, *KRT5*, expressed specifically in myoepithelial cells surrounding Ductal Carcinoma In Situ (DCIS) regions. The model trained on nine Visium breast cancer datasets was used to predict expression in a new Visium test dataset, showing highly consistent spatial patterns between predicted and observed values (Fig. 1c). Quantitatively, using the spatial autocorrelation metric Moran's I, we observed a strong positive correlation (i.e., high-high or low-

low clustering) for *COX6C* expression across most spatial spots (Fig. S2a). We applied the same model, trained on the nine fresh frozen Visium samples, to two external, previously unseen out-of-distribution FFPE samples: one measured using the version-1 FFPE protocol and the other with the FFPE CytAssist protocol (Fig. 1d). Results for one representative cancer-immune marker, *ATP1A1*, are shown for both FFPE test samples (Fig. 1d). Results for all 14 cancer-immune markers, evaluated using PCC and Moran's I metrics, demonstrate consistently positive scores, although performance varies across samples (Fig. S2c-e). This spatially-aware quantification offers a distinct advantage over non-spatial models such as HE2RNA[7], which do not provide per-tile gene expression predictions.

### Prediction performance - more robust across diverse datasets and sample types

To assess the robustness of the approach, we increased heterogeneity in the training and testing data by including samples from different protocols, laboratories, and populations. We trained a model using nine Visium datasets from fresh frozen tissues, with three datasets generated by 10x Genomics and six by Swarbrick's Laboratory[15]. Using an ensemble approach, the model trained on fresh frozen samples was applied to two external FFPE samples processed using Visium protocols. The top predicted gene, *ATP1A1*, showed high PCC values (0.51 and 0.52) and consistent gene expression patterns (Fig. 1d). Assessing model performance across 1522 functional genes (trained simultaneously), we found that predictions were positively correlated with measured values for 1136 of 1522 genes. The top 10 genes achieved PCC scores around 0.4 (Fig. 1e). Notably, model performance was consistently higher when our image normalisation strategies were applied (Fig. S3). We also thoroughly assessed the impact of image tile resolution and found that the default 299 × 299 tile size in STimage yielded the highest predictive performance (Fig. S4). These findings highlight options for improving robustness.

The trained model incorporated both the public dataset from 10x Genomics and Swarbrick's dataset, while testing was performed on an independent dataset generated by a European laboratory, consisting of 33 tissue samples. STimage was benchmarked against six other approaches: five software tools, THItoGene, IGI_DL, STnet, His2gene, and Hist2ST, and an alternative model, ViT_NB. STimage consistently outperformed all other methods across the majority of test samples, except for six (out of 33), where its performance was comparable to the top-performing models (Fig. 2e). To further enhance performance, we developed an ensemble model, FSTimage, which integrates multiple foundation-STimage models. Each model combines a foundation model as the upstream image extractor with the STimage negative binomial module. The foundation models used include GigaPath[16], H-Optimus[17], Virchow2[18], Phikon[19], CTransPath[20], CONCH[21], UNI[22], and Virchow[23]. Final FSTimage predictions were obtained by averaging results across these models. Our findings indicate that STimage, Virchow2_STimage, and FSTimage consistently outperformed all other models compared.

To further evaluate robustness and increase sample diversity, we incorporated additional datasets, including kidney cancer (fresh frozen, *n* = 615,632 spots), non-cancer liver disease (FFPE, *n* = 419,967 spots), and skin cancer (FFPE, *n* = 1014,405 spots) (Fig. S7, S6, S5). We used a leave-one-out model training and testing strategy to evaluate model performance on individual patient samples. Among the top 100 most predictable genes, Gene Ontology GO enrichment analysis revealed terms corresponding to relevant tissue biology across the three GO databases (Fig. S7a-c, S6a-c, S5a-c). The PCC between predicted and ground truth gene expression values among the top 100 genes showed a positive correlation, ranging from 0.2 to 0.8 (Fig. S7d, S6d, S5d). Performance also varied across patients, suggesting tissue heterogeneity. Representative spatial expression plots are shown in Fig. S7e-g, S6e-g, S5e-g.

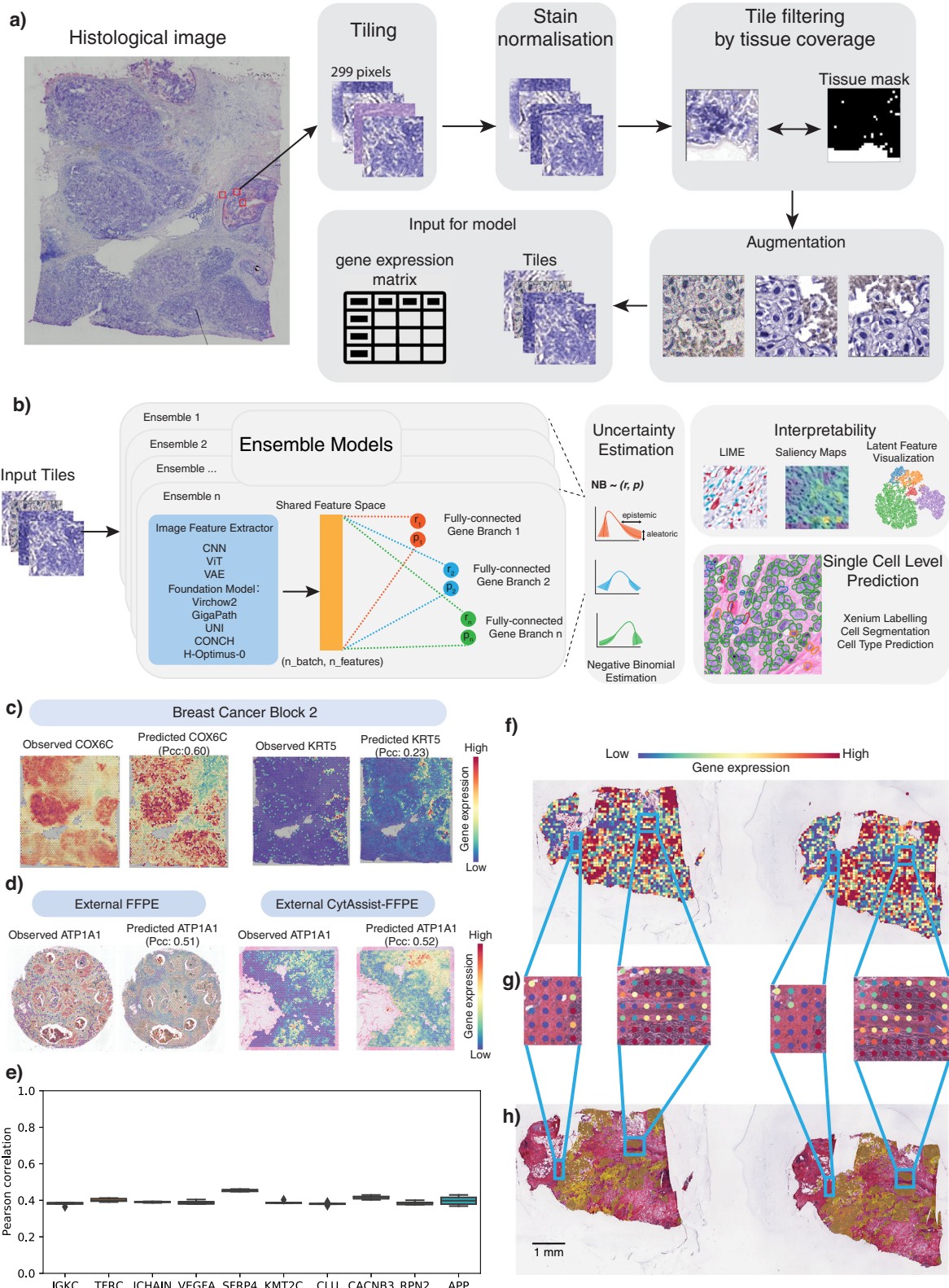

## Prediction uncertainty can be quantified for each gene

In addition to improving robustness, ensemble models allow us to quantify uncertainty arising from both the data (i.e., aleatoric uncertainty) and the model itself (i.e., epistemic uncertainty). Existing methods, such as STnet[9] and HisToGene[10] use a CNN or ViT as an image feature extractor. Both models rely on multi-layer neural networks to predict fixed gene expression values, without estimating variability in

their predictions. Hist2ST[11] employs a zero-inflated negative binomial distribution to predict gene expression distributions, but it does not account for epistemic uncertainty. To evaluate these capabilities in STimage, we used the model trained on the nine Visium fresh frozen breast cancer datasets to test its performance on two independent, unseen FFPE tissues from separate cohorts (Fig. 2a–d and S1). Two representative genes were selected for comparison: *ATP1A1*, a breast

**Fig. 1 | Overview of the robust and interpretable STimage model. a** STimage image preprocessing workflow includes image tiling, stain normalisation, quality control (QC) for tiles, and image augmentation. **b** STimage CNN-NB interpretable regression model. The STimage model consists of an image feature extractor (CNN, ViT, or foundation models) and negative binomial (NB) layers for estimating gene expression distributions. Negative log-likelihood is used as the loss function. **c** ST-measured (observed) and STimage-predicted gene expression for the highly abundant cancer marker *COX6C* and the lowly abundant but spatially distinct keratin marker *KRT5* on the test dataset. PCC between predicted values and observed values across all spots (measured by spatial transcriptomics) is shown. **d** Model performance for an out-of-distribution FFPE dataset from 10x Genomics. **e** The Pearson correlation coefficient (PCC) values for the 10 most predictable functional genes (i.e., top 10 PCC scores out of 1522 functional genes), calculated for two

external FFPE datasets using an ensemble model. The box indicates the 25th-75th percentiles with the median as a central line. Whiskers extend to the main data range (±1.5 IQR), and outliers are shown as individual points. **f–h** STimage regression model applied to non-spatial whole-slide images (WSIs) of a HER2+ patient from the TCGA dataset. Two adjacent tissue sections were used to assess reproducibility. **f** The WSI was cropped into non-overlapping tiles of size 299 × 299 pixels. A model trained on nine Visium samples was applied to predict the expression of the cancer marker gene *COX6C*. Each tile is treated as a spatial spot; red and blue colours indicate high and low predicted *COX6C* expression, respectively. **g** Two zoom-in views of tumour-enriched areas. Predicted cancer marker expression is highly correlated with tumour morphology and consistent across replicates. **h** Pathologist annotation of tumour regions (indicated in brown) in the two replicate tissue sections.

cancer marker, and *CD63*, an immune marker. For each gene, aleatoric uncertainty, epistemic uncertainty, and predicted gene expression were visualised in spatial tissue plots (Fig. 2a, b). Aleatoric uncertainty (i.e., the average of predicted NB variance values across single models) was found to be higher than epistemic uncertainty (i.e., the variance of NB mean estimates) in both FFPE samples. As expected, both uncertainty types showed a positive correlation with gene expression levels. Interestingly, lower epistemic uncertainty often coincided with higher aleatoric uncertainty, as observed for *ATP1A1* (Fig. 2b), thereby increasing confidence in the predicted NB parameters (i.e., mean and variance).

In addition to lower epistemic uncertainty, *ATP1A1* also showed higher predictive performance (PCC: 0.52; Fig. 2b) compared to *CD63* (PCC: 0.34; Fig. 2a). This finding aligns with interpretations of epistemic uncertainty, which reflects the confidence in model predictions[24]. In this case, the higher epistemic uncertainty observed for *CD63* may be partially attributed to the fact that the test samples were generated using different technological platforms compared to those in the training data. As a result, the input data were considered out-of-distribution.

## An ensemble approach improves uncertainty performance

To increase model robustness across a broad range of diverse, unseen datasets, we trained an ensemble of STimage regression models. Ensemble models have been shown to improve generalisation in the transcriptomics context[25]. We observed variation in model predictions, even when using the same training dataset and model architecture, but with different random seeds. To assess and mitigate this variability, we analysed the distribution of performance across individual models within 12 different groups and for two distinct tissue slides (a total of 30 models). The results are shown as box plots in Fig. 2c. Each box represents the average PCC from independent predictions made by 10 single models for each of 12 genes (out of a total of 14 trained genes; 2 were not presented in the additional CytAssist FFPE dataset). The lines represent ensemble models obtained by averaging predictions from all 10 single models within each group, with error bars indicating the standard deviation across different genes (12 genes). Although there is considerable variation among individual models, their performance remains consistent across ensemble groups. Furthermore, the ensemble models outperform the single models and exhibit reduced variability across all groups (Fig. 2c). We further demonstrate that this ensemble strategy also applies to the FSTimage_ens model, where embeddings from different foundation models are combined to further enhance performance (Fig. 2e).

## Out-of-distribution performance for models trained on a small dataset

To assess model performance on out-of-distribution (OOD) data, we designed an experiment where the STimage model and three other models, STnet, HisToGene, and Hist2ST, were trained on fresh frozen samples (Visium poly-A captured data) and tested on FFPE data

(Visium probe-hybridisation data). The probe-hybridisation data detected more genes and provided more information per gene. As the two datasets were derived from different tissue types and technologies, they were considered OOD (Fig. S1).

Previous studies, such as Hist2Gene[10] and Hist2ST[11], showed limited performance when trained on small datasets. In contrast, we found that the STimage model with a fine-tuning option improved performance on small training sets compared with models using ViT for feature extraction. For a robust comparison, we predicted expression levels for the top 1000 highly variable genes (HVGs) and computed both PCC and Moran's I scores for each gene. Across all 1000 genes, STimage consistently outperformed the other three models.

We further evaluated robustness by testing the prediction performance for each of the 1000 HVGs, comparing STimage to the three other methods (Fig. S1). Notably, methods based on ViT (e.g., HisToGene and Hist2ST), which generate attention maps across tissue sections, require substantially more memory. These models become challenging to run on datasets with more spatial measurements, a scenario increasingly common with recent advances in spatial transcriptomics. STimage uses a CNN-based image extractor that treats each spatial measurement as an independent data point, enabling scalability to datasets with high spatial resolution. Additionally, STimage leverages pre-trained models for feature extraction, an approach not used by ViT-based methods.

We further evaluated robustness and increased sample diversity by incorporating additional datasets, including kidney cancer (fresh frozen, $n = 6$, 15,632 spots), non-cancer liver disease (FFPE, $n = 4$, 19,967 spots), and skin cancer (FFPE, $n = 10$, 14,405 spots) (Figs. S7, S6, S5). We used a leave-one-out training and testing strategy on tissue samples to evaluate model performance for each patient. Among the top 100 most predictable genes, GO enrichment analysis revealed terms consistent with the underlying tissue biology across the three GO databases (Figs. S7a–c, S6a–c, S5a–c).

The PCC between predicted and ground truth gene expression values among the top 100 genes ranged from 0.2 to 0.8 (Figs. S7d, S6d, S5d), demonstrating a positive correlation. Performance also varied among patients, highlighting tissue heterogeneity. Example spatial gene expression plots are shown in Figs. S7e–g, S6e–g, S5e–g.

## Out-of-distribution performance on independent spatial transcriptomics datasets

In the following sections, we further evaluated the robustness of our model by testing it under more challenging scenarios, involving independent datasets from different cancer types (skin and breast cancer), diverse data types, and various experimental protocols, including low-resolution legacy ST, Visium, single-cell resolution Xenium, non-spatial TCGA images, and PhenoCycler Fusion data.

The model, initially trained on nine Visium breast cancer datasets using a functional gene list of 1522 genes, was applied to the HER2ST[26] legacy ST dataset. In this experiment, H&E images from the HER2ST

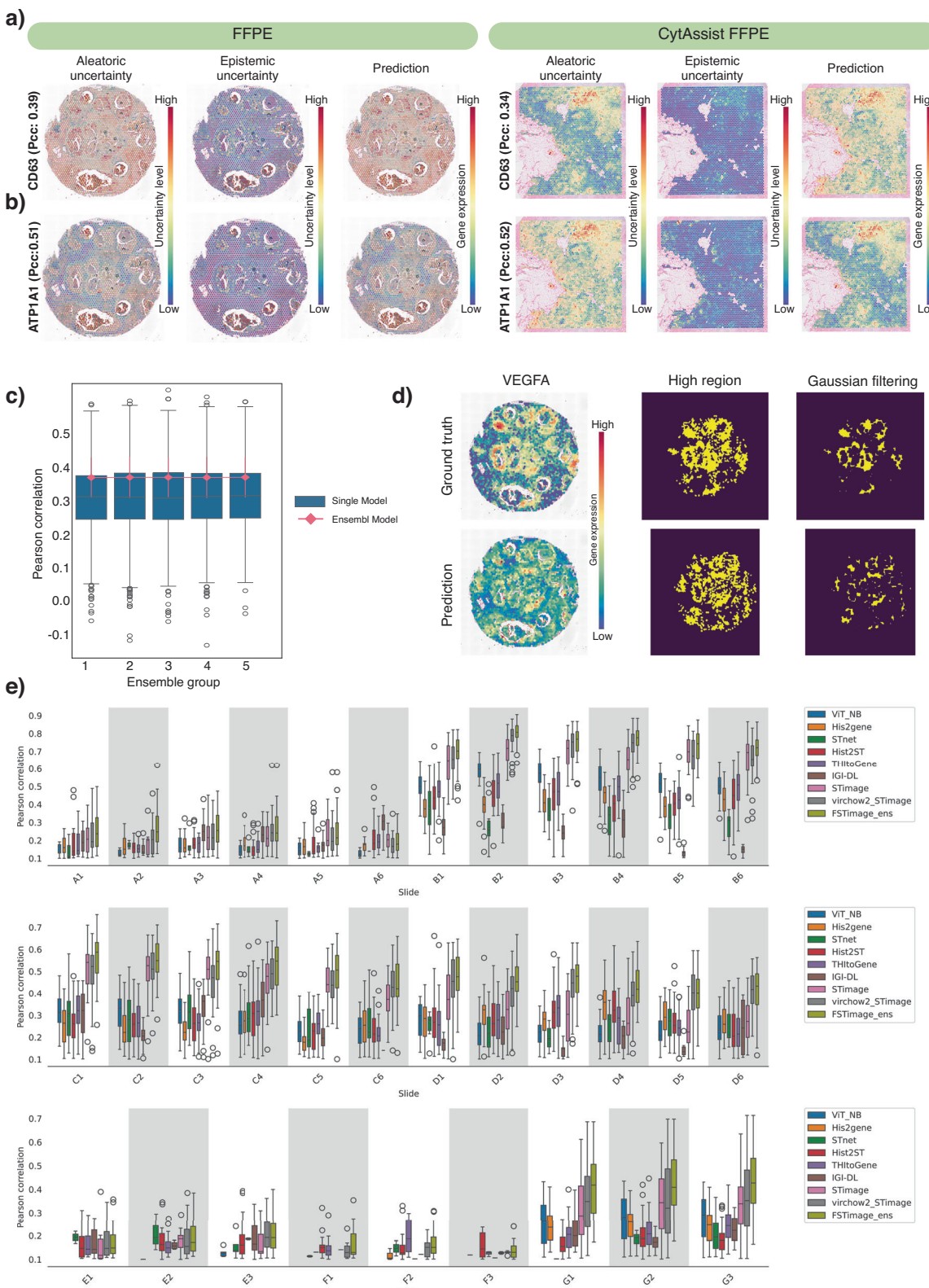

dataset were tiled at the spatial location of each spot, using the Visium spot size of 55 μm, which is smaller than the original legacy spot size of 100 μm. Gene expression predictions were generated for these smaller tiles, visualised, and evaluated against the ground truth measurements. Figure 3a displays one of the top predictable genes, *ESR1*, while additional examples, *ERBB2* and *B2M*, are presented in Fig. S8. Figure S8a and b show samples with low and high prediction performance, respectively, while Fig. S8c provides pathological annotations

for these samples. Even in low-performance predictions, we observed consistent spatial expression patterns when compared with both ground truth and pathological annotation. A quantitative assessment of a large number of genes in the HER2ST dataset is presented in Fig. S8d. This includes PCC values for 1146 predicted genes, derived by intersecting the 1522 predictable genes with the gene list available in the legacy ST data. We found that a model trained on the high-resolution Visium dataset was able to effectively predict data from the

**Fig. 2 | Distribution-based estimation and benchmarking of the STimage CNN-NB model. a** Uncertainty estimation and predicted gene expression for *CD63* in two external FFPE datasets. The red-blue colour gradient indicates high to low uncertainty levels. **b** Uncertainty estimation and predicted gene expression for the cancer gene *ATP1A1* (the top predicted gene with high PCC). **c** Performance of five ensemble models compared to 10 single models within each ensemble group. The x-axis shows five ensemble groups, each containing 10 runs. The y-axis shows the PCC between predicted and ground truth gene expression. The boxplot shows the distribution of prediction performance across 10 single models (with different random seeds) for two tissue slides per group, where boxes show the 25th–75th percentiles with the median as a central line. Whiskers extend to the main data range (±1.5 IQR), and outliers are shown as individual points. The connected lines represent ensemble models (aggregating all single models per group), with the median as a central line and error bars indicating standard deviation across genes. **d** Customised IoU score developed to better evaluate spatial model performance. **e** Comprehensive benchmarking against STnet, HisToGene, THItoGene, IGI-DL, and Hist2ST using 33 tissue slides from the HER2ST dataset[26]. The plot shows PCC values between predicted gene expression of HVGs and ground truth measurements. Each boxplot represents one test sample in the Leave-One-Out Cross-Validation (LOOCV) experiment per method. 'virchow2_STimage' refers to the STimage model using foundation model embeddings from Virchow2. 'FSTimage_ens' denotes the ensemble model using five foundation model weights from HEST. The box indicates the 25th–75th percentiles with the median as a central line. Whiskers extend to the main data range (±1.5 IQR), and outliers are shown as individual points. Source data for all comparisons are provided as a Source Data file.

legacy Visium protocol. Prediction performance was highest for samples B1-B6 (median PCC values ranging from 0.4 to 0.6), moderate for samples C1-D6, G1-G3, and H1-H3 (median PCC values from 0.2 to 0.4), and lowest for samples A1-A6 and E1-F3. Notably, samples A1-A6 and E1-F3 consistently showed low prediction performance, even in in-distribution datasets, as shown in Fig. 2e.

In contrast to the previous OOD analysis using a reduced-resolution test dataset (Visium legacy), here we applied the model to higher-resolution data from Xenium. The same model, trained on Visium H&E images, was applied to images from the single-cell resolution Xenium dataset (Figs. 3c and S9). Subcellular gene expression measured in the Xenium breast cancer dataset was grouped to match the Visium spot size, serving as ground truth for comparison with model predictions. We again observed highly consistent spatial expression patterns for gene markers such as *ESR1* (Fig. 3c), although the quantitative correlation was lower than for datasets of the same resolution (Fig. S9).

### Out-of-distribution performance on another cancer type assessed by spatial proteomics datasets

The same model, trained on nine Visium breast cancer datasets and a functional gene list of 1522 genes, was applied to H&E images from Visium skin cancer data (Figs. 3b and S10), i.e., for a different cancer type. We demonstrate that a model trained on breast cancer can predict gene marker expression in skin cancer. Predictions for key melanoma markers such as *SOX10* and *CD34* were highly consistent with protein data measured by PhenoCycler Fusion (Fig. S10). The subcellular protein staining signal from the PhenoCycler Fusion skin cancer dataset was grouped to match Visium spot size, serving as ground truth for comparison with predictions from H&E images on the same slide (Figs. 3d, S10). The ability to predict expression patterns that match protein markers highlights the model's promise for diagnostic applications.

### STimage performs well when trained and tested on other cancer types

Using the same model architecture, we trained the STimage regression model on a skin cancer dataset (Fig. S5), a kidney cancer dataset (Fig. S7), and a non-cancer liver immune disease dataset (Fig. S6). A Leave-One-Out Cross Validation (LOOCV) strategy was used to assess model performance. To further evaluate the biological applicability of the gene expression prediction model on HE images, we performed clustering on the predicted gene expression data to analyse tissue heterogeneity. Overall, the clustering results identify the structural organisation of skin layers (Fig. S18a) with expected gene markers. Notably, Cluster 1 corresponds to the melanoma region. We further utilised a set of marker genes associated with epidermal genes, dermal fibroblast genes, and sweat gland genes to characterise clusters. We found that epidermal gene markers were enriched in Cluster 0, while sweat gland genes were enriched in Cluster 5 (Fig. S18b). The spatial patterns of the clusters generally align with the H&E morphology.

Overall, STimage performed consistently well, and the top predictable genes were highly relevant to cancer biology and the tissue of origin.

### The classification model provides complementary information on tissue regions with high vs low gene expression

In addition to predicting continuous gene expression values, STimage also implements a classification model that enables the prediction of high vs low gene expression directly from H&E images (Fig. S11c). Refer to "STimage classification model" for details on the classification model. In this model, continuous gene expression values were discretised into two categories: high vs low (Fig. S11a, b).

We evaluated the classification model's performance on the top 100 predictable genes in two test samples, FFPE and 1160920F, using Moran's I, AUC, and IoU metrics (Fig. S11c). As the aim of this model is to classify tissue regions into high vs low expression zones, IoU is the most informative metric. The top predictable genes showed consistent classification performance across both test samples (Fig. S11d, e). Since the classification model predicts at the tile level, we found that it performed well on tiles with high expression, but was noisier for tiles with intermediate expression levels. Examples of top-predictable genes include *CD24*, *GNAS*, and *VEGFA* (Fig. S11d, e). These genes are highly relevant to cancer; for instance, *CD24* is known to be over-expressed in HER2+ tumours and is associated with poor prognosis[27]. To assess the classification model's robustness, we compared two genes with contrasting expression distributions, *CD52* and *CD24* (Fig. S11). *CD52* is an immune-related gene and a favourable biomarker in breast cancer[28]. Class probabilities from the model were consistent with the ground truth continuous gene expression values (Fig. S11g). All three performance metrics supported the model's predictive accuracy for both genes.

### Interpretability analysis

We investigated the interpretability of the model by assessing feature importance using LIME (Fig. S12; see Methods, "Model interpretation of STimage"). As examples, we selected *VEGFA* and *CD52*, as they were ranked among the top predictable genes and are known cancer and immune markers, respectively. We calculated LIME scores for the regression models (FFPE and 1160920F), focusing on the top five tiles with high *VEGFA* expression from cancer regions and the top five tiles with high *CD52* expression from immune regions (Fig. S12c, d). LIME scores overlaid on H&E images, used for both qualitative and quantitative assessment, are shown in Fig. S12a, b. Two representative examples are displayed for two different test datasets: FFPE (Fig. S12a) and 1160920F (Fig. S12b).

The interpretability analysis shows that different nuclei contribute differently to gene expression prediction (Fig. S12a, b). We observed that nuclei with positive LIME scores for *VEGFA* have a larger area than those with positive LIME scores for *CD52* in cancer regions, consistent with the biological expectation that cancer cells are typically larger than immune cells. When integrated with pathological annotations, we also found more nuclei with high LIME scores for

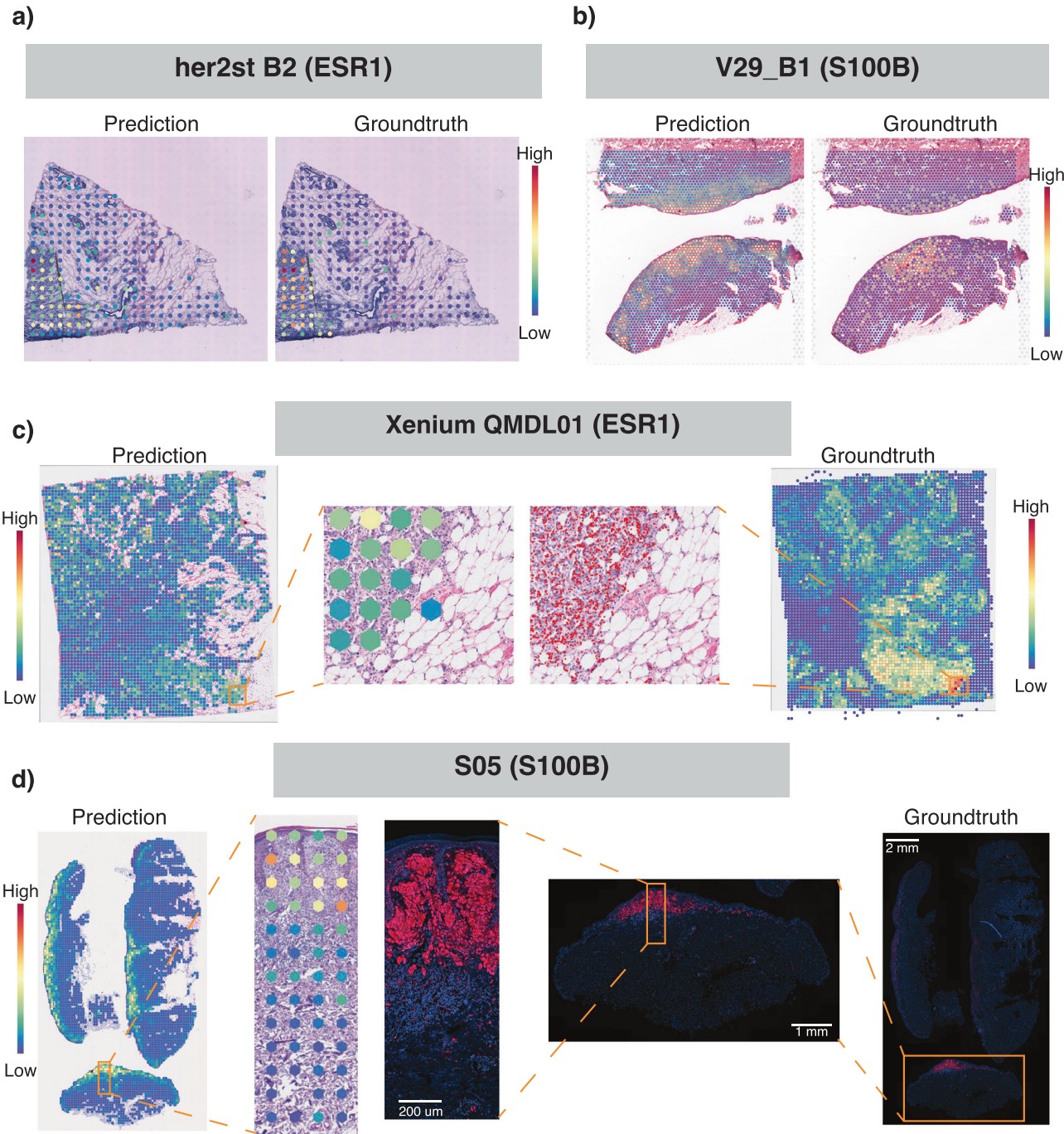

**Fig. 3 | Out-of-distribution assessment of model training on Visium breast cancer data and a functional gene list of 1522 genes, with predictions across various datasets and cancer types using different spatial technologies. a** Spatial plot of predicted and ground truth gene measurements of the breast cancer marker gene *ESR1* from legacy ST of breast cancer. **b** Spatial plot comparing predicted and ground truth gene measurements of the skin cancer marker gene *S100B* from Visium skin cancer data. **c** Spatial plot of predicted measurements of the breast cancer marker gene *ESR1* from the Xenium dataset; the middle section shows a zoomed-in comparison of spot-level predictions and single-cell resolution gene measurements. **d** Spatial plot of predicted measurements and ground truth protein measurements of the skin cancer marker gene *S100B* from PhenoCycler Fusion skin cancer data; the middle section shows a zoomed-in region showing predicted gene expression at the spot level alongside a staining image of S100B protein expression.

*VEGFA* in cancer tiles (Fig. S12a, b) than in immune tiles. This observation is consistent with the known biology that *VEGFA* is overexpressed in cancer cells in HER2+ samples[29]. To support user engagement and exploration of interpretability, we developed an interactive web application that allows visualisation of LIME scores from the STimage model on arbitrary tiles.

Moreover, analysis of the latent space revealed that the learned features from ResNet50 could effectively distinguish cancer tiles from non-cancer tiles. We overlaid the true expression and predicted probability of *CD24*, one of the top predictable genes across both datasets, on the first two principal components of the latent space (Fig. S11f). *CD24* is highly expressed in cancer regions and aligns well with pathologist annotations in Fig. S12a, b. In addition, saliency mapping confirmed that ResNet50 focuses on the most informative features (nuclei) in the images (Fig. S12).

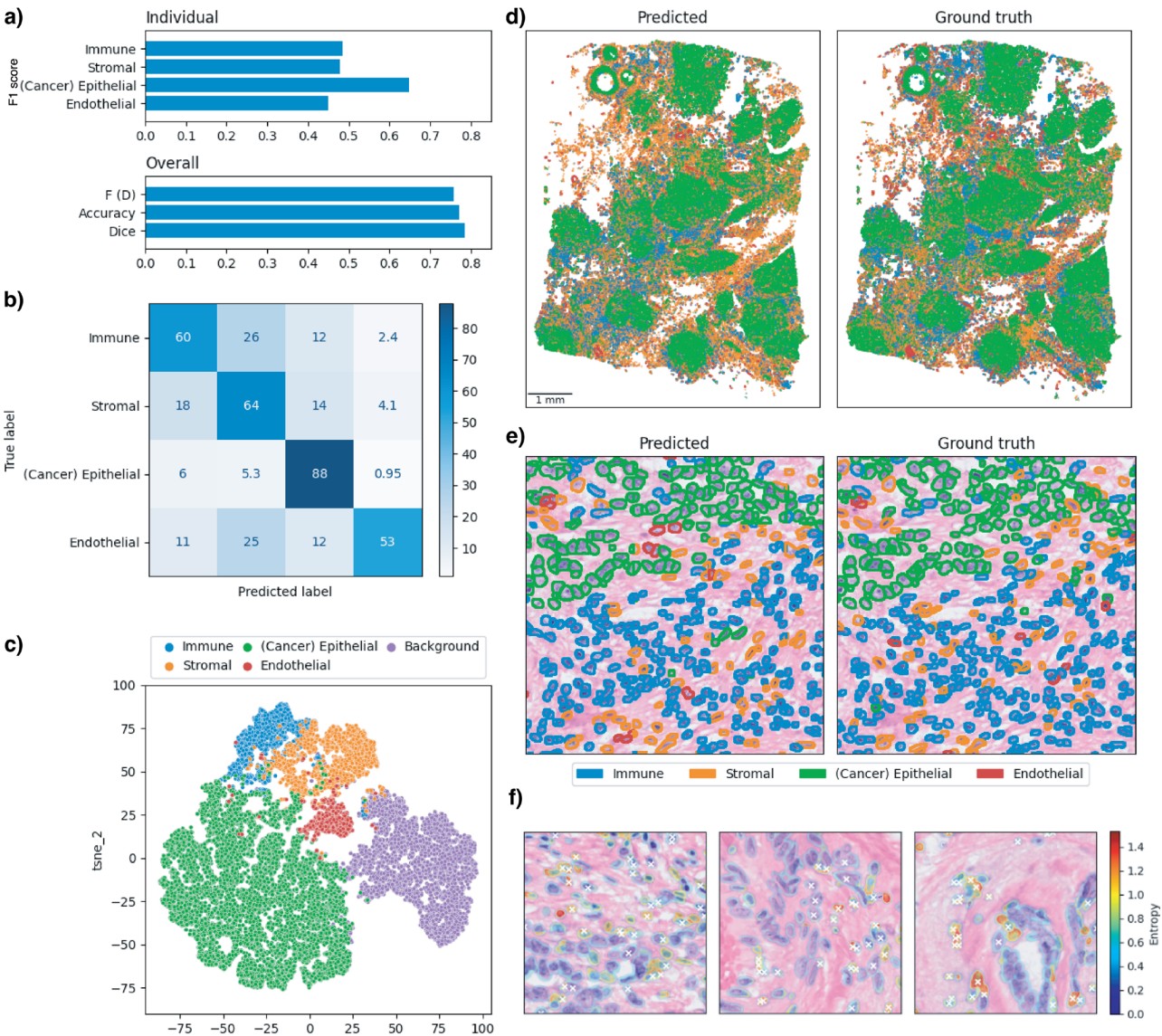

**Fig. 4 | Cell type classification at single-cell resolution using a customised Hover-Net model on Xenium data. a** Classification and segmentation metrics evaluated on the held-out test sample. The top panel shows individual classification F1 scores for each cell type. The bottom panel reports overall metrics: F (D) is the detection F1 score, accuracy refers to the classification accuracy within correctly detected instances across all cell types, and the Dice coefficient measures segmentation quality. **b** Confusion matrix for correctly detected instances, showing percentages normalised across the true labels (rows). **c** Visualisation of the features corresponding to predicted cell types using t-SNE. **d** Predicted and ground truth cell types across the full held-out test whole-slide image. **e** Close-up example of segmentation and classification of nuclei in the test image, comparing predictions to ground truth. **f** Visualisation of prediction uncertainty, measured by the entropy of predicted cell class probabilities. White crosses indicate misclassified instances.

## Xenium cell type classification

The Xenium platform from 10x Genomics enables RNA transcript detection with subcellular resolution. Compared to Visium, Xenium-generated data is particularly well-suited for deep learning-based cell classification due to its single-cell resolution profiling. We obtained Xenium data for five primary breast cancer tissue sections and trained a deep learning model to classify nuclei using whole-slide H&E images.

Figure 4 shows the classification metrics (Fig. 4a, b), indicating that cancer epithelial cells were the most accurately predicted, while other cell types also showed reasonable accuracy. Qualitatively, predictions closely matched the ground truth labels at both the macroscopic scale (Fig. 4d), where distinct tissue structures were well reconstituted, and the microscopic scale (Fig. 4e), showing individual nuclei segmentation and classification.

We also visualised the feature space learned by the model to interpret inter-cell type similarity and dissimilarity. The 256 × 256 × 64 feature map from the final layer of the classification branch was associated with the predicted cell types. Feature vectors were sampled for each cell type and the background class, and visualised using 2D t-SNE embeddings (Fig. 4c). The background was clearly separated from cells, while stromal, immune, and endothelial cells showed substantial overlap, suggesting potential classification challenges among these types.

Furthermore, we assessed model uncertainty by computing the entropy of predicted class probabilities across the image. We overlaid this uncertainty map with misclassified instances. Figure 4f shows that regions of high entropy were enriched for misclassified cells, while low-entropy regions largely corresponded to correct predictions.

We performed leave-one-out cross-validation (LOOCV) across the five Xenium datasets. Results presented in Figure S14 show consistent performance across test samples, demonstrating the robustness of our classification model. Among the cell types, cancer epithelial cells were most accurately predicted, likely due to their abundance and distinctive morphology. Immune cells (including B cells, T cells, and myeloid cells) formed the second most predictable group. This suggests potential clinical applications in profiling immune cell distribution in the tumour microenvironment, which is critical for cancer diagnosis. With more training data, particularly for rarer cell types, predictive performance could be further improved.

We benchmarked our model against the recently published CellViT model[30], a Vision Transformer-based architecture that supports both single-cell segmentation and classification (Fig. S15). At the tile level, both CellViT and the STimage classification model accurately predicted cell segments. However, the STimage model outperformed CellViT in cell type prediction (Fig. S15a). We also compared mean IoU scores across LOOCV test samples (Fig. S15b), demonstrating that our model achieved superior performance.

Notably, the STimage classification model was trained using Xenium data in which training labels were generated automatically via image registration. This process aligns the DAPI-derived cell segmentation mask with cell type annotations obtained from marker gene expression measurements in the Xenium dataset, which are considered more accurate than those inferred from H&E alone. This automated approach reduces the need for manual labelling and minimises human error.

### Evaluating clinical and translational potential of STimage using the TCGA dataset and a drug response dataset

To evaluate whether STimage could be applied to tissue images from a non-spatial dataset, we first tested two randomly selected breast cancer samples from The Cancer Genome Atlas (TCGA) (Fig. 1f–h). As ground truth for comparison, we obtained pathological annotations for these two H&E images. The predictions for cancer markers were consistent across the two TCGA replicates (Fig. 1f–h). We observed strong agreement between the predicted expression and pathological annotation, with high expression of cancer markers in tumour regions and low expression outside tumour areas.

We further demonstrate that the model-predicted gene expression can cluster cell types from H&E images, even in the absence of spatial measurements. This was evaluated using five tissue sections from three randomly selected TCGA patients. Louvain clustering based on the expression of 1522 predicted genes per tile is shown in Fig. S13a. The resulting clusters correlated with pathological annotations for each sample (Fig. S13a, b). Spatial plots of two known breast cancer markers, *ESR1* and GATA3[31], also aligned with annotated tumour and normal regions (Fig. S13c, d).

For a broader analysis, we predicted expression levels of 1522 highly variable genes for 1034 H&E images from 670 TCGA patients. The average Pearson correlation coefficient between predicted and bulk-measured gene expression was 0.48 (Fig. 5a). STimage performed consistently well for both lowly expressed genes (e.g., *MPPED1*) and highly expressed genes (e.g., *VEGFA*), as illustrated in Fig. 5b. Notably, the predicted gene expression was also informative for patient survival (see Methods, "Validation using non-spatial data and assessing predictive values for survival and drug response"). Using the top five genes identified from univariate Cox regression models, we predicted gene expression for 1034 images with matched survival data. Survival curves stratifying patients into low- and high-risk groups were significant and consistent for both true and predicted expression values (Fig. 5c, d).

Top genes with the highest PCCs from gene-wise correlation analysis for each subtype were: *FGFR3* (PCC = 0.46) and *PAK1* (PCC = 0.47) for HER2; *IL5RA* (PCC = 0.19) and *CCL13* (PCC = 0.20) for Luminal; and *KLF4* (PCC = 0.47) and *BDNF* (PCC = 0.48) for TNBC. These genes

are known to be associated with treatment resistance or breast cancer progression. For example, *PAK1* copy number is increased in HER2-positive cases[32], and its overexpression is linked to poor prognosis and metastasis[33]. *KLF4* is a known prognostic marker in TNBC[34].

For the survival benchmarking analysis, among the four comparisons, including: (1) all samples, (2) HER2-positive, (3) Luminal, and (4) TNBC, the STimage outperforms all the other four models in 1, 2, and 3, while it ranked the second best in the fourth comparison. (Fig. S16b)

To further evaluate the clinical utility of STimage, we independently assessed its ability to predict responses to drugs. Here, we analysed another cohort of breast cancer patients with treatment response information[35]. Average gene-wise Pearson's correlation coefficient was found to be 0.47 for 1139 common genes (Fig. S17b). We then evaluated the performance of true and predicted gene expression of 1139 common genes for classifying pathological complete response (pCR) vs residual disease (RD) using an radial basis function kernel support vector machine (SVM) classifier. Given the different distribution of the data, the predicted gene expression data was log-transformed, while the true raw gene expression data were CPM-normalised and standardized.

We used a two-level stratified cross-validation approach to test how well our model could predict new data. First, we divided the full dataset into five equal folds. Each time, we used four folds to train the model and the remaining fold as a held-out set. Within the four folds, we did another round of stratified 5-fold splitting to get the best-trained model to be used for predicting the 5th held-out set. We repeated this process five times so that every patient was part of the held-out set. This was done to ensure the results are robust and reliable. STimage could stratify patient responses with AUC = 0.70, compared with using the bulk RNA-seq with AUC = 0.78 (Fig. S17c).

## Discussions

While digital pathology has the potential to benefit greatly from spatial tissue analysis, it still faces significant challenges related to cost, time efficiency, reliability, and explainability[36]. We developed STimage, a comprehensive machine learning software to predict spatial gene expression and cell types accurately by training on a small set of spatial transcriptomics data, and subsequently applying the model to standard H&E images. This cost- and time-efficient approach is able to predict genes and cell types biologically relevant to the disease, while maintaining reliability across datasets and interpretability.

The STimage regression model integrates deep learning and statistical approaches to estimate the distribution of gene expression in spatial transcriptomics (ST) data and quantify prediction uncertainty. We benchmarked the STimage regression model against existing methods in both leave-one-out cross-validation experiments and out-of-distribution settings. In most cases, STimage outperformed other methods. Additionally, STimage includes a classification model that can predict high or low expression levels for a gene or gene list. STimage also implements a LIME-based interpretability model to identify meaningful tissue and cellular features that contribute to model predictions.

As spatial data remains limited due to high costs and lack of access to the spatial platforms, the datasets used in this study are modest in size. Nonetheless, we used highly information-rich data, comprising 87,166 spatial spots and 2.1 million cells from 76 patients, in addition to 670 patients from the TCGA. Despite the modest sample size, the study comprehensively covers different spatial resolutions (from single-cell to large-spot scale), protocols (polyA-capture, probe-based, and imaging-based), sample types (fresh frozen and FFPE), molecular modalities (RNA and protein), cancer types (skin, breast, and kidney cancer), and non-spatial data. This diversity enabled a robust assessment of model generalisability.

Although STimage outperformed existing tools, Pearson correlation values remain modest overall. However, spatial expression

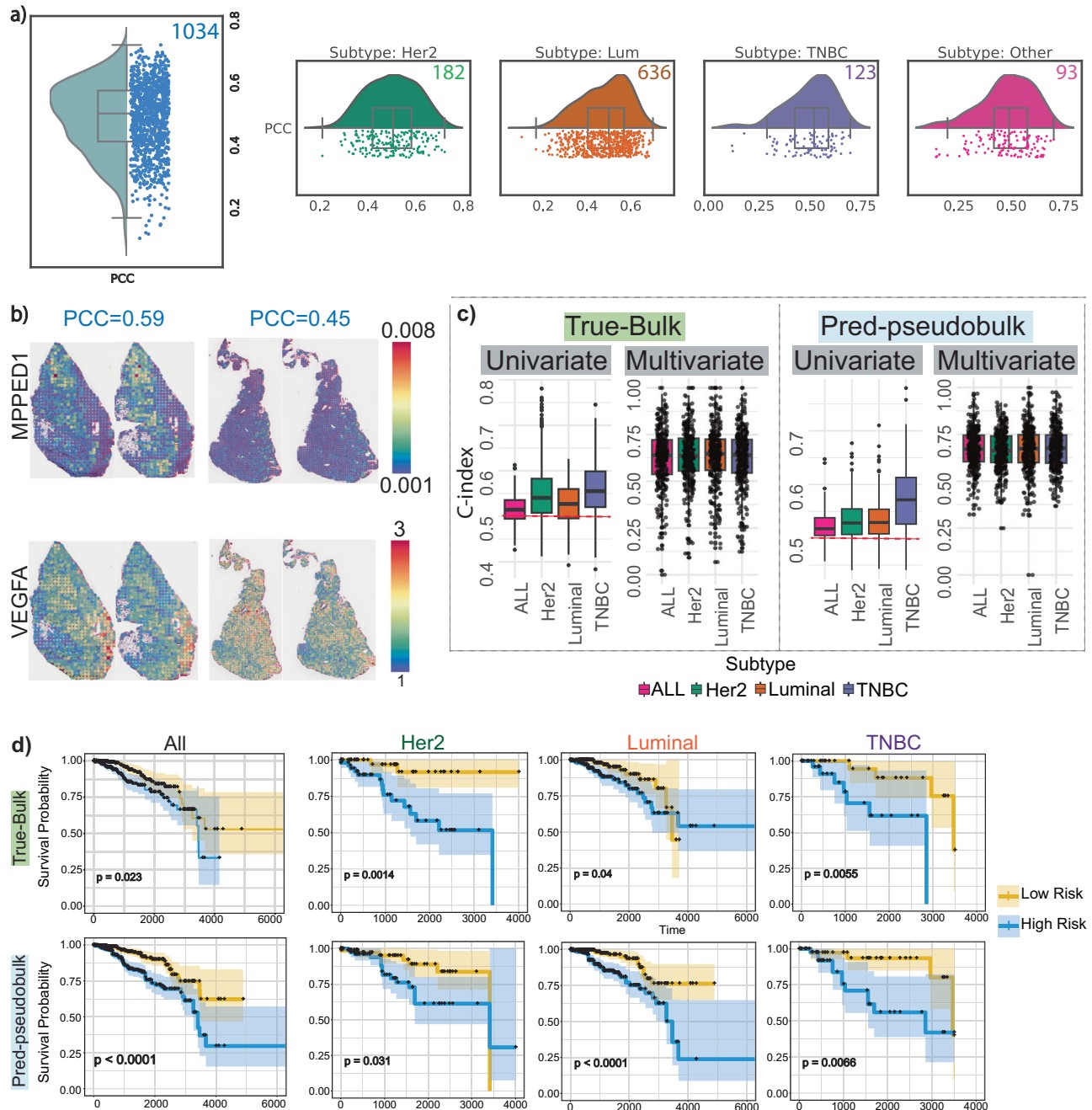

**Fig. 5 | TCGA validation. a** Box plot showing gene-wise PCC per sample, computed between predicted gene expression from TCGA H&E images and matched bulk gene expression across 1034 samples/images from 670 patients (each dot represents a patient), the box plot has median = 0.5, Q1 = 0.41, Q3 = 0.57, and whiskers at 0.167-0.722 (≤1.5 × IQR). The accompanying violin plots correspond to individual subtypes; the number in the top right of each indicates the number of samples in that subtype. **b** Spatial gene expression from the Visium dataset for a highly expressed gene (*VEGFA*) and a lowly expressed gene (*MPPED1*) for two randomly selected patients. **c** Box plots showing performance of prognostic models using the c-index for true and predicted data across univariate models (*N* = 300; each dot represents C-index score of top-300 genes) and multivariate models (*N* = 300; each dot represents C-index score of the test-set computed using 3-fold cross-validation with 100 repeats). The box indicates the 25th-75th percentiles with the median as a central line. Whiskers extend to the main data range (±1.5 IQR), and outliers are shown as individual points. Source data for all C-index scores are provided as a Source Data file. **d** Kaplan-Meier survival curves stratifying patients into low- and high-risk groups, based on true (measured bulk RNA-seq) and predicted gene expression from images for TCGA samples. The survival curves are plotted with 95% confidence interval using the log-confidence method. Multivariate models based on both true and predicted data captured significant differences between low- and high-risk groups.

patterns are arguably more biologically relevant. Therefore, we propose an evaluation metric for spatial distribution, Intersection over Union (IoU), to better reflect spatial prediction accuracy. This metric suggests the pattern prediction performance for STimage and potential utilities in clinical settings. Notably, STimage identifies the most

predictable genes, which are functionally relevant to the respective cancer types, demonstrated as relevant enriched pathways for skin cancer, lung cancer, kidney cancer, breast cancer, as well as non-cancer liver disease. For example, *SOX10* (a highly sensitive and specific nuclear marker for melanocytic lesions, including primary and

metastatic melanoma, *PRAME* (overexpressed in many melanomas but not in benign nevi or normal melanocytes, a marker and a target for immunotherapy and vaccine-based approaches in melanoma treatment), and *S100B* (a sensitive marker for melanocytic differentiation in tissue samples) were the top genes with the highest prediction accuracy for melanoma. Similarly, for breast cancer, the top predictable genes include *ESR1* (major driver in hormone receptor positive breast cancer), HER2 (*ERBB2*, a member of the EGFR family of receptor tyrosine kinases, associated with higher cancer progression and recurrence), GATA3 (strong correlation with ER-positive tumours and favourable prognosis), and immune markers like CD45 (marking immune infiltration). We assessed potential clinical utilities through survival analyses and patient stratification for drug responses. The predicted results based on the images were able to classify long vs short survival and complete vs partial responses. While interpretability measures are still in early development, initial results suggest informative latent space embeddings that can differentiate pathological annotations and identify predictive nuclear features.

Overall, we believe STimage represents meaningful progress in applying machine learning to digital pathology and provides a practical, interpretable, and extensible tool for spatial transcriptomics-based gene expression and cell type prediction.

## Methods

### Datasets

The training step requires spatial transcriptomics data as input, with large histology images cropped into 299 × 299 pixel tiles. Spatial coordinates encoded by spatial barcodes for each spot, and gene expression values for each matched spot, are used as labels to train the model. The number of microns corresponding to 299 pixels depends on the image resolution and the size of the Visium spots. The tiling process ensures that each tile contains one spot of 55 $\mu$m for 10x Visium and 100 $\mu$m for the Visium legacy. Therefore, the 299 × 299 window represents a square tile of 55 $\mu$m in dimension for the 10x Visium, and a separate model is trained for this resolution compared to a model trained for a resolution of 100 $\mu$m. With this setup, a well-trained model can be applied to histological images (non-ST data) to predict gene expression at the spot level.

First, we focused on breast cancer datasets, including an in-house Xenium single-cell spatial transcriptomics dataset and multiple publicly available breast cancer datasets that differ in resolution (33 samples from the legacy ST platform and nine samples from the 10x Visium platform) and sample preservation types, with two fresh frozen and one Formalin-Fixed Paraffin-Embedded (FFPE) sample among the nine Visium samples. To benchmark STimage, we compared it with three existing methods: STnet[9], Hist2Gene[10], and Hist2ST[11], using the 33 samples from the legacy ST platform[9]. Model training, evaluation, and testing were conducted using a leave-one-out cross-validation (LOOCV) strategy to benchmark the prediction accuracy of the regression model against the existing models.

To test the generalisability of the model to datasets generated from different laboratories and independent cohorts, we used the nine-sample Visium dataset, including six Visium samples for HER2+ breast cancer[15] and three from the 10x Genomics public dataset. The dataset was divided into 70% training data and 30% validation data. The test data included one FFPE sample from 10x Genomics and one fresh frozen sample from the HER2+ dataset[15]. For independent validation, we randomly selected three whole-slide histological images from the TCGA HER2+ cohort. In this way, the trained models from the STimage software on the Visium dataset were applied to assess their applicability to non-spatial data.

We also produced subcellular-resolution data using Xenium for five samples, and we demonstrated the capability of using this data to validate gene expression predictions as an out-of-distribution (OOD) test. We then further extended the model to predict cell types for

hundreds of thousands of individual cells. In all model training scenarios, the data were split at the sample level (WSI), not at the tile level, to avoid overfitting. We also performed OOD evaluations in several different scenarios. The regression model trained on the Visium breast cancer dataset was used to make predictions on H&E images from the legacy ST HER2+ dataset[9].

Beyond testing for breast cancer data, we also assessed STimage on two other cancer types (skin and kidney cancer datasets) and a non-cancer liver immune disease dataset (primary sclerosing cholangitis). The in-house skin cancer dataset includes 13 melanoma patient samples, processed using the Visium FFPE protocol. The kidney cancer dataset comprises six Visium samples[37], and the liver dataset includes four Visium samples[38]. LOOCV was used to evaluate the STimage model performance for each dataset separately. To evaluate cross-platform prediction, and on H&E images from in-house PhenoCycler Fusion spatial proteomics data of skin cancer, to compare the predicted gene expression with the protein modality. The detailed dataset information can be found in Supplementary Table S4

### Xenium in situ gene expression analysis

The in-house FFPE breast cancer samples were sectioned at 5 $\mu$m thickness and mounted on Xenium slides (10x Genomics). In situ gene expression profiling was performed using the Xenium In Situ Platform (10x Genomics) according to the manufacturer's protocol with the Pre-designed Xenium gene panel targeting 280 genes. In brief, tissue sections underwent deparaffinisation and decrosslinking, followed by probe hybridization overnight. The correct-binding probes were circularised by ligation and amplified by rolling cycle amplification. Autofluorescent quench and nuclei staining were performed prior to Xenium Instrument, software version 1.4.2.0. Raw fluorescence images were processed using Xenium onboard Analysis version 1.4.0.6 with default parameters for cell segmentation, transcript decoding, and assignment.

### Skin cancer Visium dataset

The in-house FFPE tissue blocks of 13 melanoma patient samples from 10 patients were sectioned at 5 $\mu$m by rotary microtome, and the sections were processed for spatial sequencing library preparation following the Visium Spatial Gene Expression for FFPE User Guide (CG000407, CG000408, CG000409 - 10x Genomics). SpaceRanger V1.0 was used to map FASTQ reads to the human reference genome (version GRCh38-3.0.0) to generate the gene expression count matrix.

### Skin cancer Phenocycler Fusion dataset

Single-cell spatial phenotyping of melanoma Formalin-Fixed Paraffin-Embedded (FFPE) slides was performed using the Phenocycler Fusion platform in collaboration with Akoya Biosciences (Marlborough, MA, USA). Tissue sections were stained in a single step with a 33-plex antibody panel. The imaging protocol was established with the Fusion Experiment Designer software according to the manufacturer's instructions. Following the acquisition of all cycles, a final composite QPTIFF image file was exported from Phenocycler Fusion software and aligned to the H&E image from the same tissue section.

### Preprocessing of the input image

Technical variation between H&E images can be caused by various factors, such as different staining procedures, imaging hardware and settings, or operators. Such technical artefacts can affect model training and testing, negatively impacting generalisability. In STimage, we perform stain normalisation for each image, such that the mean R, G, and B channel intensities of the normalised images are similar to those of the template images, while preserving the original colour distribution patterns. STimage uses StainTool V2.1.3 to perform Vahadane[39] normalisation as the default option. We show that this processing step improves model performance.

In addition, the nature of tissue sectioning will inevitably result in some tiles containing low tissue coverage. These tiles should be removed. STimage uses OpenCV2 for tissue masking and removes tiles with tissue coverage lower than 70% (default).

## STimage regression model

We utilised both sequencing data and H&E image pixel intensity data to train STimage regression models for predicting spot-level gene expression based solely on imaging data.

We adopted the NB-distributed likelihood $\mathbf{NB}(\mathbf{r}, \mathbf{p})$ for the STimage regression model, where $\mathbf{r} \in \mathbb{N}_+^G$ denotes the "number of successes", $\mathbf{p} \in (0, 1)^G$ the "probability of each trial's success", and $G$ is the number of (assumed) independent genes. The parameters of the NB distribution, $\mathbf{r}$ and $\mathbf{p}$, are estimated from the spatial spot image $\mathbf{s}$ using an image feature extractor (ResNet50 by default), $f_\theta$, with learnable pre-trained parameters $\theta$ from the ImageNet dataset. The functions $h_\omega$ and $h_\eta$ represent parameter spaces containing the weights and biases of the neural network layers that connect the latent space $\hat{\mathbf{z}}$ to the two output neurons $\mathbf{r}$ and $\mathbf{p}$.

$$\hat{\mathbf{z}} = f_\theta(\mathbf{s})$$
$$\hat{\mathbf{r}} = \text{Softplus}(h_\omega(\hat{\mathbf{z}})) \tag{1}$$

$$\hat{\mathbf{p}} = \text{Softmax}(h_\eta(\hat{\mathbf{z}})) \tag{2}$$

The original STimage regression entails the ResNet50 CNN $f_\theta$ mapping a H&E spot tile, $\mathbf{s}$, to a latent representation $\hat{\mathbf{z}} \in \mathbb{R}^{n\text{-features}}$, which is then fed into separate output layers (1) and (2), estimating $\hat{\mathbf{r}}$ and $\hat{\mathbf{p}}$, respectively. We denote estimates for the $g$-th gene's likelihood parameters by $\hat{r}_g$ and $\hat{p}_g$. Therefore, the STimage regression model defined $g$-th (random) gene expression $X_g|\mathbf{s}$, as a function of H&E spot $\mathbf{s}$, by

$$X_g|\mathbf{s} \sim \text{NB}(\mathbf{s}|\hat{r}_g, \hat{p}_g) \tag{3}$$

$$\mu_g(\mathbf{s}) = \mathbb{E}[X_g|\mathbf{s}] = \hat{r}_g(1 - \hat{p}_g)/\hat{p}_g \tag{4}$$

$$\sigma_g(\mathbf{s}) = \sqrt{\mathbb{V}[X_g|\mathbf{s}]} = \sqrt{\hat{r}_g(1 - \hat{p}_g)}/\hat{p}_g, \tag{5}$$

where $\mu_g(\mathbf{s})$ in (4) is the mean prediction for the $g$-th gene's expression with expectation operator $\mathbb{E}[.]$, and $\sigma_g(\mathbf{s})$ in (5) is the standard deviation function with variance operator $\mathbb{V}[.]$.

Overall, the ResNet50 CNN layers are optimized to take H&E images as input to estimate each gene expression's predictive distribution with NB's parameters $r_g$ and $p_g$ (for all $g \in \{1, ..., G\}$, Eq. (3)). The ResNet50 base, $f_\theta$, is used to perform convolution operations and spatial filtering to extract a three-dimensional feature map (i.e., tensor) from input image tiles. The pretrained $f_\theta$ of the ResNet50 includes the GlobalAveragePooling reduction operation[40] to convert the three-dimensional tensor to the feature vector $\mathbf{z}$, whereby each $\mathbf{z}$ corresponds to a single tile/spot within a H&E image.

The fully-connected output layers $h_\omega$ and $h_\eta$ were fine-tuned by maximising the NB's negative log-likelihood,

$$\sum_{g=1}^{G} \sum_{i=1}^{B} \log \frac{\Gamma(\hat{r}_{g,i})\Gamma(x_{g,i} + 1)}{\Gamma(\hat{r}_{g,i} + x_{g,i})} - \hat{r}_{g,i} \log(\hat{p}_{g,i}) - x_{g,i} \log(1 - \hat{p}_{g,i}), \tag{6}$$

with batch size $B$, Gamma function $\Gamma(.)$, and $x_{g,i}$ representing the observed RNA count of the $g$-th gene (i.e., gene expression).

To assess the model's predictive capability, we first trained our model on a panel of 14 marker genes, which included three immune marker genes (*CD74, CD63, CD81*) and eleven cancer-associated genes

(*COX6C, TP53, PABPC1, GNAS, B2M, SPARC, HSP90AB1, TFF3, ATP1A1, FASN, VEGFA*). Two 10x fresh frozen breast cancer samples were used (replicate 1 for training and replicate 2 for testing) to evaluate the model's performance in biomarker prediction. We then extended the model to nine breast cancer datasets, initially using the top 1000 most variable genes and subsequently applying it to a comprehensive set of the most relevant functional genes (1522 genes). The top 100 genes were selected as the most predictable based on their Pearson's correlation coefficient (PCC). During training, the model also learned the relationships between these genes.

We further improved STimage by integrating the latest foundation models for image feature extraction. The H&E image accompanying the ST data is preprocessed in the same way as described earlier. Instead of using ResNet50 as the backbone of STimage, we utilise a foundation model to convert H&E tile images into feature embeddings. These embeddings are then connected to STimage's neural network layers to estimate gene expression distributions. To demonstrate the improvement, we incorporated Virchow2, one of the most advanced foundation models, which has been trained on 3.2 million whole-slide H&E images and contains 632 million parameters. Due to its extensive training dataset and large-scale parameterisation, Virchow2 serves as a more suitable image feature extractor compared to traditional models such as ResNet50. Further, we created an ensemble model named **FSTimage_ens**, where we integrated multiple foundation models into the STimage base model. Each foundation STimage model contains one foundation model as the upstream image extractor and an STimage negative binomial module. The foundation models used are: *GigaPath, H-Optimus, Phikon, cTranspath, CONCH, UNI*, and *Virchow2*. We then aggregated by averaging the results from these models to achieve a final FSTimage ensemble (**FSTimage_ens**) prediction. We found that *STimage, Virchow2_STimage*, and **FSTimage_ens** performed better than any other models that we compared to, including *THItoGene* and *IGI_DL*.

## Uncertainty assessment

STimage estimates a NB distribution to model each gene's (random) expression $X_g \in \{0, 1, ...\}$ conditional on the H&E spot $\mathbf{s}$, i.e., $X_g|\mathbf{s} \sim \text{NB}(\hat{r}_g, \hat{p}_g)$. A single STimage model produces the distribution $p(X_g|\mathbf{s}) = \text{NB}(\mathbf{s}|\hat{r}, \hat{p})$. An ensemble of STimage models produces a distribution $p(X_g|\mathbf{s}, \Theta)$. Therefore, following the law of total variance, we can obtain two terms accounting for uncertainty:

$$\mathbb{V}[X_g|\mathbf{s}] = \mathbb{V}[\underbrace{\mathbb{E}[X_g|\mathbf{s}, \Theta]}_{\mu_g(\mathbf{s})}] + \mathbb{E}[\underbrace{\mathbb{V}[X_g|\mathbf{s}, \Theta]}_{\sigma_g(\mathbf{s})}], \tag{7}$$

where the variance of the mean estimates, $\mathbb{V}[\mu_g(\mathbf{s})]$, defines epistemic uncertainty, and the average of variances, $\mathbb{E}[\sigma_g(\mathbf{s})]$, defines aleatoric uncertainty. Aleatoric uncertainty reflects the noise inherent in the data. Epistemic uncertainty reflects the model's limitations (e.g. lack of information due to small sample sizes or suboptimal model parameters) and can be reduced with more data and/or improved models.

Applying the formula (7), we quantify total uncertainty for the prediction of each gene in each spot. Briefly, we trained 30 STimage models for approximately 1522 functional genes separately, using the same training dataset, each with a unique random seed. Corresponding values of $X_g$ were predicted for individual spots $\mathbf{s}$ using each trained model with model parameters $\Theta$. Based on the 30 individual STimage models, we computed five ensemble models. Each STimage ensemble model was derived by averaging NB estimates $\mu_g(\mathbf{s})$ from five randomly pooled STimage single models. For each of the single models and ensemble models, PCC was then calculated as the prediction accuracy metric.

## Model interpretation of STimage

STimage implements model explainability in order to highlight important segments/features (nuclei) from the image that contribute to the high expression of a gene. We used a local model-agnostic approach (LIME)[41], in which different segments of the images are repeatedly perturbed to measure how the predictions made by the trained model change, in order to score each individual segment.

Morphological diversity among cells in the tumour is an important hallmark that provides clues for histopathological assessment. In order to obtain a score for each nucleus, we used Cellpose-3 segmentation[42] to segment the nuclei from H&E images. We performed an image transformation, converting RGB images to Haematoxylin-Eosin-DAB (HED) format before applying the Cellpose segmentation. Inputs required by LIME are a trained regressor/classifier, an image tile, a gene name, and a segmentation function. Using LIME, we identified nuclei segments corresponding to high or low feature importance in the regression model. Given below are the formulas used to compute the LIME values.

$$\xi(x) = \underset{g \in G}{\mathrm{argmin}}\, L(f, g, \pi_x) + \Omega(g), \tag{8}$$

where

$$L(f, g, \pi_x) = \sum_{z, z' \in Z} \pi_x(z)(f(z) - g(z'))^2 \tag{9}$$

LIME finds a simple model $g$ that can approximate our complex neural network model $f$. The total loss in LIME is given in Eq. (8). The term $L(f, g, \pi_x)$ in (8) is a weighted sum of squared errors (SSE), where the weights are given by $\pi_x(z)$, depending on the proximity of the local neighbourhood data point $z$ to $x$. This ensures that points close to $x$ are given greater weight.

Eq. (9) creates a local approximation of our trained model in the neighbourhood of a given input image $x$ using a simple, interpretable regression model $g$, where $G$ is a set of sparse linear regression models and $\pi_x$ denotes the kernel function that defines the local neighbourhood around data point $x$. Here, $f(z)$ is the prediction made by the neural network for features $z$, and $g$ is a simple model that approximates the behaviour of $f$ in the locality of $x$, as defined by $\pi_x$. The vector $z'$ is an explainable subset of $z$. The second term in (8) is a regularisation term applied to $g$, which assigns zero weight to non-important input features.

**STimage Web Application.** To provide an interactive, visual interpretability option, we have developed the STimage Web App, where users can select image tiles and genes of interest to visualise the nuclei important for the prediction of gene expression. The Web App allows users to either use a pre-trained model or upload their own trained model for interpretability visualisation. We provide trained STimage regression models for breast cancer, kidney cancer, liver disease, and skin cancer. The Web App can be accessed at: https://gml-stimage-web-app.streamlit.app/.

## STimage classification model

We reasoned that categorical prediction of gene expression, mapping whether genes are present or absent, or highly or lowly expressed in specific tissue regions, could already have significant clinical potential.

$$z = \frac{x - \mu}{\sigma}. \tag{10}$$

We transformed the raw gene expression values (continuous) to $z$-scores using (10), and categorised them as "Low" for $z$-scores below zero and "High" for $z$-scores above zero. For the classification model,

we used a pre-trained ResNet50 (optionally with a fine-tuning layer) as a feature extractor for all tiles, and then applied a regularised logistic regression classifier, optimising the log-loss (binary cross-entropy) function.

$$\widehat{y_i} = \frac{1}{1 + \exp -(\beta' X + b)}, i \in \{1, \ldots, N\}, \tag{11}$$

We used the "elastic net"[43] to balance model performance and complexity by applying both L1 and L2 penalties. In Eq. (11), the coefficient set $\boldsymbol{\beta} = (\beta_1, \ldots, \beta_n)$ and $\boldsymbol{X} = (x_1, \ldots, x_n)$ denote the latent space features ($N = 2048$ features).

$$L_{logistic}(\beta) = -\frac{1}{N}\sum_{i=1}^{N} y_i \log(p_{y_i}) + \sum_{i=1}^{N}(1 - y_i)\log(1 - p_{y_i}), \tag{12}$$

$$L_{enet}(\beta) = L_{logistic}(\beta) + \omega\left((1 - \alpha)\sum_{j=1}^{2048}\beta_j^2 + \alpha\sum_{j=1}^{2048}\left|\beta_j\right|\right) \tag{13}$$

The elastic loss $L_{enet}$ in Eq. (13) is the sum of the binary cross-entropy loss (log loss) from Eq. (12) and the regularisation terms.

Here, $\alpha$ in Eq. (12) is a mixing parameter defining the relative contribution of RIDGE and LASSO to the penalty term. For $\alpha = 1$, the ElasticNet function reduces to LASSO regularisation (-logistic loss + sum of absolute values of weights), while for $\alpha = 0$, the function reduces to RIDGE regularisation (-logistic loss + sum of squares of weights). $\omega$ is the regularisation parameter, and we set the regularisation strength to $\omega = 0.1$ (a smaller value implies weaker regularisation). The solver "SAGA"[44] was used for optimisation, as it is faster for large datasets.

The classification model was trained on seven breast cancer samples for the top 100 highly predictable breast cancer genes (from the STimage regression model) and tested on 10x FFPE and fresh frozen HER2+ (1160920F) samples.

## Benchmarking and performance assessment

We applied a LOOCV strategy on 33 samples (from a total of 36, with 3 excluded due to low quality) from the HER2+ ST dataset to evaluate STimage regression model performance and benchmark it against existing methods such as STnet[9], HisToGene[10], and Hist2ST[11]. Briefly, all models were evaluated using the same data-splitting LOOCV strategy, where models were trained on 32 samples, and the spatial gene expression was predicted on the held-out sample.

For each gene, Pearson's correlation coefficient (PCC) was calculated between the predicted and experimentally measured spatial gene expression values. PCC was used as the primary performance metric to assess prediction accuracy. We also computed the autocorrelation Moran's I index as an additional metric, a weighted correlation measure that incorporates the local spatial context.

To assess generalisation, we evaluated model performance on the top 1000 highly variable genes (HVGs). After removing genes expressed in fewer than 10% of total spots, 737 HVGs were retained for further analysis. Each gene received a score from applying the model to a patient, resulting in 737 scores per patient for comparison across models.

To ensure fair comparisons, both the HisToGene[10] and Hist2ST[11] models were run using the authors' recommended hyperparameters. However, STnet failed to execute due to technical issues in the original source code, so we re-implemented the STnet model according to the original paper[9]. Once all predictions from all models were collected, we obtained 97,284 values (33 models × 4 methods × 737 genes) for comparative analysis.

We also benchmarked STimage against the other methods using an out-of-distribution (OOD) test dataset. In this scenario, all models

were trained on two fresh frozen breast cancer tissue samples from 10x Genomics and tested on an FFPE tissue sample, which we considered as an OOD sample due to the substantial differences in data generation protocols between these two sample types, poly-A capture for fresh frozen versus probe-capture for FFPE protocols. Using the same gene filtering approach as described above, we obtained 982 of the top 1000 HVGs in both the training and prediction datasets.

Hist2ST and HisToGene include a vision transformer (ViT) component as part of their neural networks, which requires loading all spot tile images and expression values from a single tissue into the model in one batch. Consequently, running these models requires greater computational resources (particularly memory) when applied to Visium data, which contain approximately 10 times more data than legacy ST datasets. Training of Hist2ST and HisToGene failed on a computing cluster equipped with an NVIDIA Tesla V100 SXM2 32 GB GPU, necessitating a workaround: we split the Visium tissue into multiple smaller subsets using a sliding window approach to reduce memory load.

Each subset contained a square tissue region of $25 \times 25$ spots with no overlap. Each subset was treated as an individual dataset, which reduced the number of spots loaded per batch while preserving the spatial neighbourhood structure. All subsets from the two fresh frozen samples were used to train Hist2ST and HisToGene models using their default hyperparameters. The trained models were then applied to predict gene expression in the FFPE dataset. Model performance was assessed using both Pearson's correlation coefficient (PCC) and Moran's I index.

## A geometric metric to better assess model predictions of spatial patterns

The commonly used metric in gene expression prediction tasks, PCC, is known to be influenced by outlier noise and is unable to capture spatial patterns. To address this, we developed a metric for evaluating model performance in terms of how well predictions match ground truth spatially. For a given gene, we standardise the predicted and ground truth values to z-scores and then categorise them as binary values, based on whether the values are greater or less than zero. These binary values are then organised into a spatial matrix, creating a binary image where gene expression is plotted according to spot coordinates. To minimise noise and retain reliable prediction patterns, we apply Gaussian filtering with a sigma value of 1 to both rows (spatial $X$) and columns (spatial $Y$). After obtaining the refined binarised prediction and ground truth images, we compute the Intersection over Union (IoU) scores to assess the overlap between the spatial patterns of the ground truth and the prediction. This score effectively reflects the accuracy of predicting spatial expression patterns.

## Cell type classification model for Xenium single-cell ST data

We modified the Hover-Net model[45], a CNN-based instance classification and segmentation model, to enable cell segmentation and cell type prediction for Xenium spatial transcriptomics data at single-cell resolution. First, we generated cell type labels by annotating the cells in the Xenium data with individual cell types using Seurat reference-based integration with the human breast cancer atlas from ref. 15. We merged similar cell type labels into four categories: immune cells, stromal cells, cancer epithelial cells, and endothelial cells. The Xenium labels were then mapped to the H&E image by applying Scale-Invariant Feature Transform (SIFT) registration[46] between the Xenium DAPI channel and the H&E image. This enabled us to obtain, for each tissue sample, segmentation labels with unique cell types for nearly every cell in the H&E images (i.e., 1,294,600 cells).

To generate the training and test data, the H&E images were divided into $256 \times 256$-pixel tiles, and tiles containing fewer than two nuclei were filtered out. The model was trained using four of the five breast cancer tissue samples and evaluated on one held-out test

sample. We also calculated the mean IoU score for each LOOCV test sample. We compared our model with the recent CellViT model[30], a vision transformer-based model that was trained to predict single-cell segmentation and cell type classification.

## Validation using non-spatial data and assessing predictive values for survival and drug responses

To perform external validation, assess robustness, and demonstrate the clinical applicability of STimage, we applied the model to the TCGA dataset. For this analysis, H&E images were downloaded from TCIA[47], and bulk gene expression and clinical data for corresponding patients were downloaded from https://www.cancer.gov/ccg/research/genome-sequencing/tcga. We obtained 3066 images from TCIA. A total of 1532 images without matched bulk and clinical data were excluded from the analysis, and the remaining 1534 images were processed for quality control using HistoQC. We filtered out low-quality images using the following parameters: blurry_removed_percent≥0.9, small_tissue_removed_num_regions > 97. 5th percentile, and pixels_to_use < 5th percentile. These thresholds were chosen based on the empirical distribution of quality parameters, removing extreme values. Using the mentioned filtering criteria, 500 poor-quality images were excluded. We finally used 1034 breast cancer images from 670 patients and predicted the gene expression of 1522 predictable genes. Predicted gene expression for each image was averaged across all spots to compare with the ground truth TCGA bulk gene expression. We measured the performance of STimage by calculating Pearson Correlation (PCC) between the average predicted expression across all the spots with the true bulk gene expression for the matched samples. We also computed the performance of STimage for predicting top-300 predictable genes for three subtypes (HER2+, Luminal and TNBC). Patient subtypes were defined according to the criteria described in ref. 48.

STimage predictions were further used to perform survival analysis and stratify patients into low- or high-risk groups. We first built univariate Cox regression models for the top 300 predictable genes using the function 'coxph' from the R package `survival`[49]. Cox regression models were built separately for true and predicted gene expression. Days to death and days to last follow-up (for patients still alive) were used as survival times, and survival status (alive/dead) was used as the event indicator. The univariate models were evaluated using the concordance index (c-index) for both predicted and true survival outcomes.

The top five genes with the highest c-index were used to build multivariate Cox regression models. We used a 3-fold cross-validation strategy with 100 repetitions to evaluate the multivariate models. The median survival risk score from the multivariate model was used to stratify patients into low- or high-risk groups. Univariate and multivariate models were constructed separately for each subtype. For both bulk and predicted data, the average c-index for the multivariate models was found to be over 0.6 across all subtypes. Model performance was assessed using the 'concordance' function from the `survival` package. Univariate Cox regression models and Kaplan-Meier survival curves were computed using the `survival` package. Multivariate models were constructed using the R package `ClassifyR`[50].

Moreover, we also compared STimage's survival performance with the existing gene expression prediction models on TCGA dataset. To conduct the survival benchmarking analysis, we utilised the publicly available code from https://zenodo.org/records/14602489[51] and integrated the results of STimage into it. To ensure fair benchmarking, for each method four-fold cross-validation strategy was used to select the best model trained on the Her2ST spatial transcriptomics dataset for the top 1000 HVGs (737 after QC for (gene less than 1000 spots)).

To further assess the clinical utilities, we added to our previous survival analysis an independent assessment of the ability to predict responses to drugs. Here, we analysed a cohort of 168 breast cancer patients treated with neoadjuvant and chemotherapy with ($n = 65$) or

without HER2-targeted therapy[35]. 161 patients were assessed at surgery using the residual cancer burden (RCB) classification. 42 (26%) had a complete response (pCR), 25 (16%) had a good response, 65 (40%) had a moderate response and 29 (18%) had extensive residual disease. We evaluated the performance of classification models using true or predicted gene expression for classifying pCR vs RD using an RBF kernel support vector machine. We used 1139 genes in the functional gene list that were present in this data. The continuous predicted gene expression data were log-transformed, while the raw count gene expression data were CPM-normalised and standardized. We applied a nested 5-fold stratified cross-validation: each outer fold used four folds for training and one for testing, with an inner 5-fold split for model selection. This process ensured every patient was tested once. AUC was calculated for each fold, and the average AUC was reported as the final performance metric.

### Ethical Approval

The data generation complied with all relevant ethical regulations and was approved by the University of Queensland's Human Research Ethics Committees (ethics approval numbers 2018000165 and 2017000318) and by the Metro South Human Research Ethics Committee (11QPAH477) and the Royal Women's Hospital Human Research Ethics Committee (2005/HE000785). Written informed consent was obtained from all participants, and no compensation was provided. Detailed patient information is provided in Supplementary Tables S1, S2, and S3. Sex was determined based on data assigned in participants' electronic medical records.

### Reporting summary

Further information on research design is available in the Nature Portfolio Reporting Summary linked to this article.

## Data availability

The datasets that support the findings of this study are available from several sources. Publicly available datasets used include Her2ST [https://github.com/almaan/her2st], Swarbrick's Lab [https://zenodo.org/records/4739739], Public breast cancer 10X-Visium [https://www.10xgenomics.com/datasets], Liver Visium (GSE240429) [https://www.ncbi.nlm.nih.gov/geo/query/acc.cgi?acc=GSE240429], and Kidney cancer Visium (E-MTAB-12767) [https://www.ebi.ac.uk/biostudies/ArrayExpress/studies/E-MTAB-12767]. A collected and processed compilation of these public datasets is accessible via UQ eSpace [https://doi.org/10.48610/4fb74a9]. Furthermore, TCGA H&E images (TCGA-BRCA) used for model inference are available from the GDC Data Portal [https://portal.gdc.cancer.gov/projects/TCGA-BRCA]. The in-house datasets generated for this study have been deposited in the UQ eSpace repository and are available upon request at [https://doi.org/10.48610/e8426d2]. These include: (1) breast cancer Xenium (raw data and H&E image); (2) melanoma Visium (raw sequencing and processed data); and (3) melanoma PhenoCycler Fusion (raw images and processed data). The raw sequencing data of melanoma Visium data have been deposited in the European Genome-phenome Archive (EGA) and is accessible under the accession number EGAD50000002172. Access to these in-house datasets is restricted due to ethical considerations and patient consent limitations. Researchers may request access via the UQ eSpace repository. Access will be granted to researchers who agree to the data use and ethics terms. Requests are typically reviewed within 24 hours, and once approved, access will be provided for a period of 12 months. Source data are provided with this paper.

## Code availability

The STimage software and web-app tools are available on GitHub at https://github.com/BiomedicalMachineLearning/STimage. This software is licensed under the BSD 3-Clause License.

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

## Acknowledgements

We thank Prof. Alex Swarbrick for kindly providing the spatial transcriptomics data. We also thank the staff at the University of Queensland Sequencing Facility and the School of Biomedical Sciences Imaging Facility for their support with spatial transcriptomics sequencing. We thank the patients and their families for consenting to the use of their tissues for research through the Brisbane Breast Bank and other tissue resources. This research was partially supported by the Australian Research Council through the Industrial Transformation Training Centre for Information Resilience (IC200100022) (Q.H.N.). Q.H.N. is supported by the Australian Research Council (ARC DECRA Grant DE190100116), the National Health and Medical Research Council (NHMRC Project Grant 2001514), and the NHMRC Investigator Grant (GNT2008928).

## Author contributions

Q.N. and X.T. conceived the project. Q.N., X.T., O.M., J.X. and S.M. designed the algorithm. S.M. led the uncertainty quantification analysis. X.T. and O.M. developed the software. C.Z., Z.X., A.M.R. and S.M.T. generated data. T.K. and P.T.S. performed pathological annotations. F.R., M.T., K.K. and N.Y. contributed to the data analysis. All authors contributed to the writing of the manuscript.

## Competing interests

M.T. and S.M. were employed by Max Kelsen, a commercial company with an embedded research team. The remaining authors declare no competing interests.
