## [Peer Review File · Nature Communications]

Robust and interpretable prediction of gene markers and cell types from spatial transcriptomics data

Corresponding Author: Professor Quan Nguyen

Version 0:

Reviewer comments:

Reviewer #1

(Remarks to the Author)

Tan et al. used very few training and test datasets (training: $n=7$; test: $n=2$) to build a weak model for BRCA, aiming to predict the spatial expression distribution of a series of genes in WSI at the spot level. The logic of the whole manuscript is confusing, and there are the following problems:

1. The selection of training and test dataset:

a) All samples were derived from only one cancer type: BRCA, and were from the 10X Visium platform. Because of the differences in histological characteristics among cancer types, does this model only predict the gene expression distribution of BRCA WSI samples? The sequencing depth of different sequencing technologies is different, and the lack of ST platform data in the training dataset will cause the problem of domain bias.

b) The number of train/test datasets is only 9, so it is necessary to increase the number of train datasets for establishing prediction models.

2. In the Materials and Methods section, the Datasets section mentioned that "The model applied to this dataset predicts 14 cancer and immune markers and corresponds to all results". In the STimage regression model, the author indicated that "We trained our model on a panel of 14 marker genes which include four immune marker genes (CD74, CD24, CD63, CD81) and ten cancer genes (COX6C, TP53, PABPC1, GNAS, B2M, SPARC, HSP90AB1, TFF3, ATP1A1, FASN)". However, in the legend of Fig. 2e, the author pointed: "Pearson correlation and Moran's I score of predicted 1000 HVGs against ground truth for each model were compared", given the above, there are the following questions:

a) Does the CNN-NB model predict the expression distribution of the 14 marker genes listed in the text or the expression of 1000 HVGs?

b) Why select these 14 marker genes? They are neither classical immunological marker genes nor BRCA-characteristic marker genes, was the selection of these 14 marker genes supported by the papers related to their function in BRCA progression?

c) Why were PCC of only 12 target genes detected in Figure 1e instead of the 14 listed above? Among them, TTLL12 is not present in the 14 marker genes listed above. The authors need .

3. In the Preprocessing of the image input section:

a) The author's description is unclear, only indicating the tools of stain normalization and filtering, without clarifying the tiling method.

b) The authors repeatedly emphasized in the article that the GEP distribution prediction is spot level, how many microns correspond to 299 pixels? Is it the spot (55um) level of 10x Visium? Moreover, this paper involves both 10X Visium and ST data with different spatial resolution data. Can 299 pixels also represent the spot level for ST data?

c) In Fig S3, the author selected several tile sizes of 224, 299, 450, 600, and 900 to try to explain the reasons for selecting 299 pixels, but firstly, the selection of these sizes is not continuous, and secondly, the spacing among tile sizes is different, with a gap of 75 between 224 and 299, 151 between 299 and 450, 200 between 600 and 900, it is recommended to start with a smaller value and do tile size scale tests every 75 intervals.

4. Figure 1c-d uses the RNA expression of 10X Visium data as the ground truth, which cannot truly reflect the spatial gene expression. It is recommended to use IF/IHC staining to correlate fluorescence intensity with GEP. For example, the 10X Visium BRCA-IDC dataset, or the IHC staining picture in the TCGA database.

5. Genes compared with "ground truth" in the manuscript are COX6C, KRT5, ATP1A1, S100A14, and C3... S100A14 and C3 are not in the 14 marker genes mentioned above. The comparison results between the predicted values of all 14 marker genes mentioned above and the ground truth are not presented in this manuscript, so it is difficult to avoid the suspicion that the suitable genes are selected for display, which means that the model cannot well predict the spatial distribution of all 14 marker genes. It is recommended that a comparison of ground truth fluorescence intensity at the protein level for all 14 marker genes be added to the supplementary figure.

6. In the STImage regression model section of the method, the authors claim that "Our assessment found that the ZINB-based model did not perform as well as the NB-based model". And "the Poisson likelihood was also deemed problematic for spatial sequencing data, which is typically over-scattered and sparse." In both cases, the authors do not provide data or cite literature to prove their accuracy.

Minor

1. In the Materials and Methods, Datasets section, paragraph 3, line 9, the PPFE sample should be FFPE sample.
2. Materials and methods part, the first paragraph in the Uncertainty assessment section, line 1, Negative Binomial (NB), lack of space. The full name of the Negative Binomial has already appeared once before, there should be an abbreviation.
3. Fig1 f-h: Fig f-g lacks the annotation of gene name, h lacks the pathologist annotation of tumor/normal region, and all feature plots lack the scale bar.

Reviewer #2

(Remarks to the Author)

Summary:

Spatial transcriptomics is a technique that can predict gene expressions from H&E slides while preserving tissue anatomy. However, this approach is costly, highlighting the necessity of developing an AI-based alternative. Current deep learning methods for this task have limitations in terms of interpretability, robustness, and performance, which are essential for clinical applications. To address this issue, the authors developed a deep-learning probabilistic framework that overcomes these limitations, providing improved robustness and interpretability, even in cases where the training datasets are limited. The developed method's performance was measured against existing methods and found to be superior.

Concerns:

The authors use LIME and SHAP to analyze which regions were important to the predictions. They state that nuclei regions contributed more to the predictions, but this visual analysis was only done for two images. To make such a statement, this visual analysis needs to be done for many more images. Also, this analysis isn't quantitative, so it's subject to interpretation and confirmation bias. I recommend removing that sentence altogether since two images aren't enough to claim this.

The caption of Fig 1g states that the predictions are highly correlated with tumor morphology, which again, requires many more samples to make such a statement. Recommend removing.

Since the interpretability aspect is a core contribution of the paper, it requires more thorough evaluation. Evaluations of explanation faithfulness need to be performed and reported. This may be added as a supplementary table or figure.

Since no statistical tests were performed, it is difficult to assess the reliability of the results. Where appropriate, statistical tests need to be conducted and their p-values should be reported.

Fig 2f is hard to read. It should be split into two rows for better readability.

The text could be worded better and contains many spelling and grammar errors, making it difficult to read. Should be put through a grammar checker tool.

Recommendation:

The method appears to improve upon the currently existing methods in terms of predictive performance. It was also tested on different protocols, labs, and populations. Therefore, I recommend accepting after the concerns are resolved.

Reviewer #3

(Remarks to the Author)

In this study the authors present a new algorithm STImage which uses machine learning to predict gene expression on H&E images after learning the spatial gene expression patterns from spatial transcriptomics data. STImage and its performance are demonstrated on breast cancer datasets.

The concept is not novel and despite seemingly good performance of STImage, there are from my point of view many parts that are immature analytically which are accompanied with several typos and inconsistency through which make the manuscript difficult to assess.

MAIN CONCERNS

- The algorithm is tested only on a very limited breast cancer datasets of breast cancer. The authors should clearly state which data are used for training and testing throughout the manuscript to enable easier comparisons between different sections.
- There are different ST-seq dataset used across the manuscript Visium 10X, Visium-FFPE and Visium legacy. While the use of the larger dataset which : Visium legacy is mentioned in the abstract and material and method, it is not seen in the results. The authors should provide more information on how the different ST-seq datasets were used in the study and consider building a training and testing dataset that includes a mix of the three methodologies to better evaluate the algorithm's performance on different types of data.
- Building on the previous comment in the material and methods the authors claim to use 1 FFPE tissue sample and when coming to the results they report testing 2 tissue FFPE samples. Such inconsistencies across the different sections in terms of how the datasets are used and reported highlight the immaturity of the manuscript.
- The authors should provide a clearer description of the output of the algorithm, both in the abstract and throughout the manuscript, to ensure that readers understand what the algorithm does and what the results mean.
- The definition of the 14 marker genes used in the study could be clarified. It is unclear how these genes were selected and whether they represent the top genes predicted by STImage. Additionally, the choice of these genes is not well-justified, and it is not clear why no stromal or endothelial markers were included. Providing more context on the selection criteria and the rationale behind these choices would help readers better understand the approach. Moreover, it could be useful to consider incorporating stromal and endothelial markers as these compartments are important in breast tumors and can also be annotated by pathologists. This would improve the algorithm's ability to predict gene expression in all compartments of breast tumors.
- The Swarbrick's lab dataset has been annotated in the original publication, could this be used for validation, interpretation here?
- The non spatial tile labelling and testing is not convincing, first it is not clearly explained how the training was performed. Second testing is performed only on two samples which are closely related since they are successive sections from the same tumor. One needs to really extend the testing to prove applicability.
- The preprocessing, processing use of the ST-seq data is not described.
- It is unclear what is the impact of OpenCV2 on tiles removal , especially the tiles situated in the edges that may contain less than 70% tissue coverage, it seems to me that some tiles in the 'middle' of the tissue with bad coverage may be removed, but also tiles on the edges or tiles containing adipose or less dense tissue? How many tiles are removed per image and where? Are important tiles removed? Is it necessary to remove them? Are these tiles creating so much noise that they need to be removed?
- Stain normalization seems to be performed to look like the RGB intensity of a template image. What is the influence of the choice of the template image. How was the template image chosen?
- The uncertainties are assessed spatially. Have the authors assessed if the uncertainties seem to be mainly driven by spatial location or the genes assessed? In other words, what is the highest source of uncertainty, the quality of the image? Or the capacity of the algorithm to predict a specific gene expression?
- Breast cancer is a very heterogeneous disease, when assessing only 14 genes across the manuscript it may be that the performances are different from subtype to subtype. No track of the clinicopathological features of the sample used is given or hint towards the model being associated with subtypes is given. Some have shown that subtype can be predicted from H&E images, it is therefore important to assess how this can affect the model.
- The interpretability analysis in the end of the 'prediction performance' paragraph is disappointing, the only feature which the author mention in their interpretability section is nuclei which has been reported before, the author attempt to conclude on other feature in a gene-specific manner... perhaps... but I am not managing to understand the last sentence which involve COX6C and KRT5 genes. It would be interesting to understand further which feature drive the prediction of different genes
- Can uncertainties by training on different ST-seq platform/ methodologies be assessed?
- What is the scalability of STImage, can more genes be assessed? It is important to provide the performance of STImage for all genes tested at least the 14 mention all along ideally all genes.
- The interpretability, last chapter, is not clear, suddenly in this chapter a new gene for immune marker appear C3 is it a typo? do the author mean CD63? or the C3 gene is used, C3 is not a typically marker gene for immune cells.

MINOR comments

- No p values are shown for the PCC (pearson correlation)
- Typo: for independend validation (p2)
- Typo: PPFE sample (p2)
- Typo: A classification model can be added top produce complementary information (p8)

Version 1:

Reviewer comments:

Reviewer #1

(Remarks to the Author)

Although the author has done more analysis on the original version, the performance of the model needs to be further improved. Some similar works in the past two years seem to have better performance than STImage, such as THltoGene and IGI_DL. The following are the additional comments for the author's modification:

1. The process of selecting the variable genes and 1,522 functional genes is unclear. Please provide a clear explanation of how these genes were chosen, including the specific methods and thresholds utilized. The rationale for selecting the highly variable genes (HVGs) and the genes used for prognosis prediction should be clearly articulated.
2. The rationale for selecting only 450 samples for the prognosis prediction analysis, rather than using the full TCGA cohort, is unclear. If any sample selection was performed, the criteria and justification must be provided. Please present the results for all TCGA samples.
3. The correlation analysis presented was conducted on a per-sample basis. Please provide the results of the correlation between the predicted gene values and the true bulk gene expression values for each gene. Are the genes with a high correlation to bulk values the ones that your gene prediction model predicts with greater accuracy?
4. The presence of outliers in the data may affect the accuracy of the Pearson correlation coefficient. The Spearman correlation coefficient results require further explanation to interpret result significance in the context of your study.
5. Moran's I is used to analyzing the spatial distribution pattern of a single variable. However, the author uses it to assess the accuracy of gene predictions without comparing it to the Moran index of actual gene values, which is confusing.
6. The details of the prognosis prediction model are missing, particularly the method used to derive the survival score from the predicted gene expression values. More thorough explanation is needed.
7. The poorer performance of the STImage model in some cases of the Her2ST LOOCV benchmark is concerning. Whether STImage is biased to the spatial pattern of a specific slice?
8. While multiple models were trained using different datasets, the benchmark analysis was limited to the Her2ST dataset and 785 genes instead of 1522 genes that used in prediction. This raises questions about the ability to accurately reflect the models' predictive performance and generalization capabilities.
9. The authors only benchmarked HisTogene, STnet, and Hist2ST released in 2021 or 2022. How do the more recently released THltoGene and IGI_DL perform?
10. The median PCC of BRCA FFPE ST data predicted by Stimage is less than 0.1 in Fig. 2E, this suggested a very weak correlation.
11. In Figure S8, the PCCs around different slides show quite different results, with some slides showing median PCCs below 0.
12. IoU values in Fig.S11g are abnormally high. From the figure, it seems there aren't many overlapping areas between prediction and ground truth.
13. The section on Xenium cell type classification does not present the results of the leave-out test comprehensively. It lacks comparative results with other classification models and evaluation metrics, such as mIoU.
14. Is there a cell type marker in the genes predicted by the model? This can be compared with the classification results.
15. The authors didn't assess prognosis performance for STImage, such as C-index.

(Remarks on code availability)

Reviewer #3

(Remarks to the Author)

Thank you to the authors for addressing the questions and comments previously raised. Although I appreciate the technical efforts to improve robustness, reproducibility, and interpretability, the manuscript remains highly technical without demonstrating the biological application of the developed tool suite. The tool suite's applicability appears complex and not easily accessible to researchers.

The main concerns are the lack of easy access and the absence of user-friendly software, as well as insufficient biological interpretations, which limit the study's interest. While the authors highlighted key novel aspects and technical improvements in their response, these improvements should be benchmarked against other algorithms. More critically, there are no new biological findings that enhance our understanding of diseases or physiological processes, which raises serious questions about the usefulness of the presented tool suite and algorithm.

In the response, the authors mentioned inter and intra-tumor heterogeneity of cancer. However, this aspect is critically missing in the manuscript. There are no measurements of intra-tumor heterogeneity or its association with gene expression and/or clinically relevant parameters. The algorithm's ability to detect phenotypically different tumor clones is neither assessed nor validated. Intra-tumor heterogeneity could be compared to matched WGS for cancer cell heterogeneity or scRNA-seq data for immune cell heterogeneity.

Additionally, with the few samples analyzed for each disease, it is probably not relevant to discuss inter-tumor heterogeneity. The authors also mention "highly relevant predictable genes for each cancer type," but it is unclear in the manuscript what these genes predict or how they reveal relevant biological insights.

In short, while I appreciate the technical improvements and the additional data used in the revised version, the manuscript still lacks critical elements such as cell type specificity, clinically relevant analysis, spatial ecosystem generalization, and specific spatial gene expression. These missing aspects result in a lack of novel biological insights, which limits the overall interest and impact of the study.

(Remarks on code availability)

On github, but could be more clearly organized

Reviewer #4

(Remarks to the Author)

All comments of the reviewer 2 has been addressed.

(Remarks on code availability)

Version 2:

Reviewer comments:

Reviewer #1

(Remarks to the Author)

The authors have improved the study, the reviewer still has following comments based on their revised manuscript.

1. The authors suggested the relatively low Pearson correlation coefficients (PCC) observed in several benchmark datasets due to insufficient training sample size. However, they proceed to perform downstream analyses using these potentially inaccurate predictions. As the task of inferring gene expression levels from H&E images is inherently a regression problem, the use of Intersection over Union (IoU) — which binarizes continuous gene expression values — may not appropriately reflect the accuracy of regression outputs. It is strongly recommended that the authors report additional regression metrics such as Mean Absolute Error (MAE) or Mean Squared Error (MSE) to provide a more comprehensive and quantitative assessment of prediction accuracy.

2. This study predominantly highlights genes with strong prediction performance, which appear to coincide with spatial patterns of H&E staining intensity. This raises the concern of potential model bias, specifically, whether STimage disproportionately favors genes with highly localized expression patterns that align with histological contrast. It would be important for the authors to clarify whether STimage struggles with genes exhibiting broadly distributed expression patterns. Furthermore, is there a correspondence between high Pearson correlation coefficients and high spatial overlap of predicted versus true gene expression domains? For genes with low PCC, is inaccuracy primarily due to incorrect prediction of spatial distribution or due to inaccurate expression value estimation?

3. Multiple previous models incorporate spatial relationships among tissue tiles either through architectural design or as part of the input features, yet STimage appears to omit such explicit spatial encoding. Despite this, the model achieves better predictive performance across multiple benchmarks. It would be valuable for the authors to discuss the implications of this design choice. Does the model implicitly capture spatial dependencies through other means, or is such information not critical for the tasks addressed?

4. This study lacks a direct comparison between STimage's survival prediction performance and existing H&E-based models, such as IGIDL survival model. The authors primarily compare their own univariate and multivariate models, which limits the reader's ability to contextualize the improvements achieved. Furthermore, the rationale for performing subtype-specific survival modeling in breast cancer cohorts is not clearly articulated. Would a pan-subtype survival model (without stratifying by subtype) result in inferior performance, and if so, could this be quantified and discussed?

(Remarks on code availability)

Reviewer #3

(Remarks to the Author)

I appreciate the thorough responses provided by the authors addressing my concerns regarding the biological applications and user accessibility of the STimage tool suite.

Overall, I am satisfied with the responses and the revisions made by the authors.

(Remarks on code availability)

The code is easier to navigate on github

Version 3:

Reviewer comments:

Reviewer #1

(Remarks to the Author)

The authors have significantly improved the manuscript. The authors did not perform a comparison between the STimage and IGI-DL, but they provided explanations for this decision. I am satisfied with the responses and the revisions made by the authors. Congrats.

(Remarks on code availability)

Reviewer #1 (Remarks to the Author): Expert in spatial transcriptomics analysis and method development, and cancer computational genomics

Tan et al. used very few training and test datasets (training: n=7; test: n = 2) to build a weak model for BRCA, aiming to predict the spatial expression distribution of a series of genes in WSI at the spot level. The logic of the whole manuscript is confusing, and there are the following problems:

We appreciate the reviewer's comment. In our original submission, we utilized the largest and most diverse collection of data available, which included HER2+ low-resolution samples (n=23 patients, 36 tissues) and three datasets with higher-resolution Visium data (n=6 patients, n=2 for Fresh Frozen samples, and n=1 for an FFPE sample). Our results demonstrated that our model not only outperformed three existing models in accuracy but also introduced significant improvements in robustness and interpretability, which were the primary focus of our work.

We observed that with the rich content provided by spatial transcriptomics data, a neural network can be effectively trained on a relatively small dataset. For instance, the STnet model (He et al., 2020, *Nature Biomedical Engineering*) was developed using the same 36 samples, comprising a total of 13,620 spots (legacy ST). In spatial transcriptomics, each sample includes thousands of individual spots, each corresponding to an image tile with thousands of gene expression labels. In our original submission, the nine Visium samples spanned three datasets: 16,456 spots (Wu et al.), 7,785 spots (10x Genomics, Fresh Frozen), and 2,518 spots (10x Genomics FFPE).

In this revised version, to further evaluate robustness and increase sample sizes, we incorporated additional datasets, including kidney cancer (Fresh Frozen, n=6, 15,632 spots), non-cancer liver disease (FFPE, n=4, 19,967 spots), and skin cancer (FFPE, n=10, 14,405 spots). Additionally, we applied our model to the latest subcellular resolution Xenium spatial transcriptomics data (n=5). This subcellular data enabled us to predict cell types at single-cell resolution and assess the generalization of the model trained on low-resolution data when applied to single-cell spatial data. Furthermore, we validated our models using a non-spatial TCGA dataset (n=450). To ensure robustness and validity, we compared our predictions with spatial proteomics data (CODEX). We believe our work represents the most comprehensive assessment of different spatial datasets for this type of model.

We apologize for any lack of clarity in the logical flow of the manuscript. In response to the reviewer's concerns, we have extensively revised the paper as detailed point-by-point below. This work addresses a significant challenge in digital pathology by leveraging the rich content of spatial transcriptomics data while tackling the high cost barrier of this technology. We developed machine learning models that use H&E image inputs to predict gene marker expression and cell types, both crucial tasks in digital pathology. Unlike existing methods, our approach focuses on

two key aspects often overlooked: robustness and interpretability. We demonstrate that our model performs well across various sample types, disease types, resolutions, and diverse cohorts, including non-spatial data. Robustness was achieved by predicting data distribution, estimating uncertainty, and employing an ensemble approach, while interpretability was ensured through feature attribution methods (LIME), latent space explanation, and integration with pathological annotations and the biological functions of the genes. Additionally, we applied our model to breast cancer survival prediction, showcasing its potential clinical benefits.

1. The selection of training and test dataset:

a) All samples were derived from only one cancer type: BRCA, and were from the 10X Visium platform. Because of the differences in histological characteristics among cancer types, does this model only predict the gene expression distribution of BRCA WSI samples? The sequencing depth of different sequencing technologies is different, and the lack of ST platform data in the training dataset will cause the problem of domain bias.

We appreciate the reviewer's question. Addressing potential domain bias is a key focus of this work, and we have thoroughly evaluated the model's performance on both in-distribution and out-of-distribution samples. Our model was trained using various protocols, including the 10x Visium poly-A capture protocol (offering comprehensive gene coverage), the probe-capture protocol (more sensitive), the legacy ST protocol (lower resolution), and the latest Xenium platform (single-cell resolution). The 10x spatial platform is unique in generating both H&E images and gene expression data by default, which is crucial for training imaging-to-expression models. Additionally, we tested the model trained on the spatial BRCA dataset by predicting gene expression in non-spatial H&E images from the TCGA dataset (n=5).

In the revision, we extended our model's evaluation to multiple cancer types, including kidney cancer (Fresh Frozen, n=6, 15,632 spots), non-cancer liver disease (FFPE, n=4, 19,967 spots), and skin cancer (FFPE, n=10, 14,405 spots). The extensive analyses have been added to the Supplementary Figures S6, S7, S8.

We also conducted a comprehensive assessment of the model's robustness using multiple training and testing strategies. Overall, we found no significant domain bias when the ST platform was excluded from the training dataset. In addition to the previously applied leave-one-out strategies, we evaluated the model on both in-distribution and out-of-distribution datasets, considering variations in resolution and different cancer types. We further analyzed which genes were most predictable by the model.

b) The number of train/test datasets is only 9, so it is necessary to increase the number of train datasets for establishing prediction models.

As mentioned above, we observed that with the rich content provided by spatial transcriptomics data, a neural network can be effectively trained on a relatively small dataset. For instance, the STnet model (He et al., 2020, *Nature Biomedical Engineering*) was developed using the same

36 samples, comprising a total of 13,620 spots (legacy ST). In spatial transcriptomics, each sample includes thousands of individual spots, each corresponding to an image tile with thousands of gene expression labels. In our original submission, the nine Visium samples spanned three datasets: 16,456 spots (Wu et al.), 7,785 spots (10x Genomics, Fresh Frozen), and 2,518 spots (10x Genomics FFPE).

In this revised version, to further evaluate robustness and increase sample sizes, we incorporated additional datasets, including kidney cancer (Fresh Frozen, n=6, 15,632 spots), non-cancer liver disease (FFPE, n=4, 19,967 spots), and skin cancer (FFPE, n=10, 14,405 spots). Additionally, we applied our model to the latest subcellular resolution Xenium spatial transcriptomics data (n=5). This subcellular data enabled us to predict cell types at single-cell resolution and assess the generalization of the model trained on low-resolution data when applied to single-cell spatial data. Furthermore, we validated our models using a non-spatial TCGA dataset (n=450). To ensure robustness and validity, we compared our predictions with spatial proteomics data (CODEX). We believe our work represents the most comprehensive assessment of different spatial datasets for this type of model.

2. In the Materials and Methods section, the Datasets section mentioned that "The model applied to this dataset predicts 14 cancer and immune markers and corresponds to all results". In the STImage regression model, the author indicated that "We trained our model on a panel of 14 marker genes which include four immune marker genes (CD74, CD24, CD63, CD81) and ten cancer genes (COX6C, TP53, PABPC1, GNAS, B2M, SPARC, HSP90AB1, TFF3, ATP1A1, FASN)". However, in the legend of Fig. 2e, the author pointed: "Pearson correlation and Moran's I score of predicted 1000 HVGs against ground truth for each model were compared ", given the above, there are the following questions:

a) Does the CNN-NB model predict the expression distribution of the 14 marker genes listed in the text or the expression of 1000 HVGs?

We apologize for the lack of clarity. We trained two models separately: one with 1,000 HVGs and another with the 14 genes. This has been clarified in the current version. Additionally, we included an analysis of the most predictable genes among approximately 1,522 biologically relevant genes, representing cell type markers, immuno-oncology genes, and others. Our findings suggest that this approach can identify highly predictable gene sets that are biologically related to the cancer types and the organs of origin.

b) Why select these 14 marker genes? They are neither classical immunological marker genes nor BRCA-characteristic marker genes, was the selection of these 14 marker genes supported by the papers related to their function in BRCA progression?

We selected the 14 markers to demonstrate the model's performance on a panel of genes, providing a relevant analysis for exploring potential clinical applications of predicting gene panels rather than individual gene markers. These genes include a set of cancer and immune marker genes identified as spatially variable in the Visium training dataset.

Specifically, the genes are COX6C (cancer metabolic processes, such as the Warburg effect in cancer glycolysis), SPARC (extracellular matrix regulation, associated with cancer metastasis and progression), HSP90AB1 (association with angiogenesis and proliferation, a potential therapeutic target), TFF3 (associated with breast cancer proliferation, hormone regulation and survival), ATP1A1 (cell proliferation, survival, adhesion, migration, and interaction with the tumor microenvironment in breast cancer), CD74 (enhance invasiveness and interact with macrophage migration inhibitory factor) and CD81 (involved in cell adhesion, migration, and signal transduction, cancer proliferation, migration and metastasis). The remaining genes are CD24 (regulate cell adhesion and migration; high CD24 is associated with increased metastasis, poorer prognosis), CD63 (may act as both tumour promoter or suppressor depending on the context), TP53 (a well established cancer marker), PABPC1 (globally manage mRNA, may contribute to uncontrolled cell growth), GNAS (contribute to cell signaling and metastasis), B2M (essential for MHC-I, can act as either tumour suppressor or promoter), FASN (upregulated in breast cancer, associated with cancer metabolism, implicated in resistance to cancer chemotherapy).

It is important to note that the purpose of testing this gene set is to illustrate the performance of the model with a gene panel, while STimage can be trained on any gene set. In the revised version, we conducted a thorough analysis of a large set of 1,522 known gene markers, including those related to cell types, immune pathways, oncology pathways, and other essential pathological processes. This approach contrasts with the common analysis of the 1,000 most variable genes, many of which may not be relevant to cancer. Based on the prediction results, we selected the genes with the highest predictability, as indicated by their Pearson correlation scores. These top predictive genes appear to be specific to cancer types and are enriched for the organs from which the cancer originates, such as kidney cancer.

c) Why were PCC of only 12 target genes detected in Figure 1e instead of the 14 listed above? Among them, TTLL12 is not present in the 14 marker genes listed above. The authors need . We apologize for the inconsistency as among the 14 markers, two genes were not present in the training datasets. In our revised version, we have extended the analyses to thousands of genes, beyond the initial 14 markers. The 14 genes include seven known markers (COX6C, SPARC, HSP90AB1, TFF3, ATP1A1, CD74, and CD81) and seven genes used in STnet (CD24, CD63, TP53, PABPC1, GNAS, B2M, and FASN) (He et al., 2020). However, since two of these genes could not be removed from the test dataset, we replaced them with two new genes, including the TTLL12. In the revised version, we have shifted our focus from the top 14 marker genes to the hundreds of most predictable and functionally relevant genes.

3. In the Preprocessing of the image input section:

a) The author's description is unclear, only indicating the tools of stain normalization and filtering, without clarifying the tiling method.

We apologize for not describing in detail the tiling method. Briefly, the spatial x and y coordinates for each spot were used to determine the center of each tile. Depending on the WSI resolution, the cropping size uses pixels/ μm to convert the pixel values for 100 μm (legacy ST)

and 55 μm (Visium). The cropped tiles were then resized to 299x299 to ensure that tiles from different samples with different resolutions have comparable area coverage. We have added this description to the method session.

b) The authors repeatedly emphasized in the article that the GEP distribution prediction is spot level, how many microns correspond to 299 pixels? Is it the spot (55um) level of 10x Visium? Moreover, this paper involves both 10X Visium and ST data with different spatial resolution data. Can 299 pixels also represent the spot level for ST data?

As described in the session above, the number of microns corresponding to 299 pixels are dependent on the image resolution and the size of the Visium spots. The tiling process ensures that each tile contains one spot of 55um for 10x Visium and 100um for the Visium legacy. Therefore, the 299x299 window represents a spot of 55 micron, a squared tile of 55 micron in dimension; a model was trained for this resolution separately from another model that was trained for a resolution of 100 micron.

As described above, the number of microns corresponding to 299 pixels depends on the image resolution and the size of the Visium spots. The tiling process ensures that each tile contains one spot: 55 μm for 10x Visium and 100 μm for Visium legacy. Thus, a 299x299 pixel window represents a 55 μm or 100 μm square tile. A model was trained specifically for a 55 μm resolution, separate from another model trained for the 100 μm resolution.

c) In Fig S3, the author selected several tile sizes of 224, 299, 450, 600, and 900 to try to explain the reasons for selecting 299 pixels, but firstly, the selection of these sizes is not continuous, and secondly, the spacing among tile sizes is different, with a gap of 75 between 224 and 299, 151 between 299 and 450, 200 between 600 and 900, it is recommended to start with a smaller value and do tile size scale tests every 75 intervals.

We thank the reviewer for the suggestion. It is noted that 224 and 299 are two commonly used image input sizes for neural networks. As suggested by the reviewer, we have added a 750 tile size to ensure 150 intervals from 450 to 900. There is no change in our conclusions and it remains that 299 x 299 is still the best. We have added the results to the Supplementary Figure 4 (Supplementary Figure S4).

4. Figure 1c-d uses the RNA expression of 10X Visium data as the ground truth, which cannot truly reflect the spatial gene expression. It is recommended to use IF/IHC staining to correlate fluorescence intensity with GEP. For example, the 10X Visium BRCA-IDC dataset, or the IHC staining picture in the TCGA database.

Our model is designed to predict gene expression, not protein expression. Therefore, the ground truth for our predictions is the gene expression values obtained from experimental spatial transcriptomics technology. Other ground truths considered in this work include bulk gene expression data (for TCGA data) and pathological annotations for comparing cell type clustering results based on predicted gene expression values with those based on pathological

annotations. We evaluated the clustering results using predictions versus the original spatial transcriptomics data, relative to the annotation ground truth (Supplementary Figure S5). Furthermore, we utilized the extensive non-spatial H&E data from the TCGA database, where bulk RNA-seq data and survival data serve as the ground truth (Figure 1d, Figure 5, Supplementary Figure S9).

Following the reviewer's suggestions, we also tested a case where we used protein expression from CODEX data as a "ground truth" to compare to predicted gene expression values based on H&E of the same section (Supplementary Figure S11). We wish to emphasize that our models are to predict gene expression, not protein expression. It is established from the literature that, most of the time, the correlation between these two modalities is mostly not high. Therefore, we expected that the performance will be lower for the proteins that are not well correlated with gene expression. Our results, nevertheless, show high consistency between two modalities. We added the following figure to the manuscript (Also shown below).

Following the reviewer's suggestions, we also tested a case where we compared predicted gene expression values based on H&E images with protein expression from CODEX data as a "ground truth" (Supplementary Figure S11). We emphasize that our models are designed to predict gene expression, not protein expression. Literature indicates that the correlation between these two modalities is often low (Pearson correlation ranges from 0.53 to 0.71, PMID:27951527). Therefore, we anticipated that performance might be lower for proteins not well correlated with gene expression. However, our results demonstrate high consistency between the two modalities. We have included this additional figure in the manuscript (also shown below).

Supplementary Figure S11.

5. Genes compared with "ground truth" in the manuscript are COX6C, KRT5, ATP1A1, S100A14, and C3... S100A14 and C3 are not in the 14 marker genes mentioned above. The comparison results between the predicted values of all 14 marker genes mentioned above and the ground truth are not presented in this manuscript, so it is difficult to avoid the suspicion that the suitable genes are selected for display, which means that the model cannot well predict the spatial distribution of all 14 marker genes. It is recommended that a comparison of ground truth fluorescence intensity at the protein level for all 14 marker genes be added to the supplementary figure.

As mentioned in response to comment #4, the use of 14 gene markers was intended to demonstrate how the model performs with a gene panel, similar to those used in current sequencing-based diagnostics. The selection of these 14 genes was based on literature rather than our experiments. In the revised version, we moved beyond the 14 genes and expanded our analysis to 1,522 well-curated functional genes. We present the performance of all 1,000 highly variable genes (Figure 1, Figure S1) and the top 100 most predictable genes (Figure S1).

For the task of predicting gene RNA expression, the ground truth is the measured RNA levels, not protein expression. Previous research indicates that protein levels are generally not highly correlated with RNA levels, with Pearson correlations ranging from 0.53 to 0.71 (PMID:27951527). Nonetheless, we assessed the predicted RNA expression against measured protein expression levels using spatial proteomics CODEX data. These new results are included in Figure 3d and Figure S11. Additionally, we have added multiple independent evaluation analyses to assess model performance. For example, we compared predictions with out-of-distribution expression data for all 106 shared genes in single-cell resolution Xenium data (Figure S10) and for all 1,146 shared genes in lower-resolution legacy spatial transcriptomics data (Figure S9). We also compared predictions for all 1,522 genes with bulk RNA sequencing data from 450 patients in the TCGA dataset and with independent pathological annotations (Figure 5).

6. In the STImage regression model section of the method, the authors claim that "Our assessment found that the ZINB-based model did not perform as well as the NB-based model". And "the Poisson likelihood was also deemed problematic for spatial sequencing data, which is typically over-scattered and sparse." In both cases, the authors do not provide data or cite literature to prove their accuracy.

We provided analysis results for the ZINB-based model performance here. We have also added results for Poisson likelihood for comparison.

Minor

We thank the reviewer again for the comments and suggestions.

1. In the Materials and Methods, Datasets section, paragraph 3, line 9, the PPFE sample should be FFPE sample.

Amended

2. Materials and methods part, the first paragraph in the Uncertainty assessment section, line 1, Negative Binomial (NB), lack of space. The full name of the Negative Binomial has already appeared once before, there should be an abbreviation.

Amended

3. Fig1 f-h: Fig f-g lacks the annotation of gene name, h lacks the pathologist annotation of tumor/normal region, and all feature plots lack the scale bar.

Amended

Reviewer #2 (Remarks to the Author): Expert in digital pathology and artificial intelligence

Summary:

Spatial transcriptomics is a technique that can predict gene expressions from H&E slides while preserving tissue anatomy. However, this approach is costly, highlighting the necessity of developing an AI-based alternative. Current deep learning methods for this task have limitations in terms of interpretability, robustness, and performance, which are essential for clinical applications. To address this issue, the authors developed a deep-learning probabilistic framework that overcomes these limitations, providing improved robustness and interpretability, even in cases where the training datasets are limited. The developed method's performance was measured against existing methods and found to be superior.

Concerns:

1) The authors use LIME and SHAP to analyze which regions were important to the predictions. They state that nuclei regions contributed more to the predictions, but this visual analysis was only done for two images. To make such a statement, this visual analysis needs to be done for many more images. Also, this analysis isn't quantitative, so it's subject to interpretation and confirmation bias. I recommend removing that sentence altogether since two images aren't enough to claim this.

We thank the reviewer for the positive recommendation. We have removed the sentence (referring to LIME and SHAP) and edited the text discussing the interpretability.

We have significantly enhanced our interpretability analyses. We first assessed interpretability at the level of precisely segmented nuclei within the context of histopathological annotation. We hypothesized that nuclei most explanatory of cancer or immune genes would exhibit high LIME scores in cancer core and stromal regions, respectively. Results supporting this hypothesis are shown in the new Supplementary Figures S12 and S13. Additionally, we observed that cancer

nuclei in cancer regions with high expression of cancer genes had a larger segmented area compared to nuclei with high LIME scores for immune genes.

Furthermore, we included interpretability analysis of the latent space (Supplementary Figure S12f) and assessed consistency with pathological annotations (Supplementary Figures S13).

An important aspect of our model's explainability is predicting the biological signatures underlying the predictions, such as known gene markers and cell types. To address this, we evaluated the interpretability of the top predictable genes concerning the signatures of cancer types and organ types (new Supplementary Figures S6, S7, S8). We found that the most predictable genes are enriched for signatures of these cancer and organ types.

2) The caption of Fig 1g states that the predictions are highly correlated with tumor morphology, which again, requires many more samples to make such a statement. Recommend removing. Since the interpretability aspect is a core contribution of the paper, it requires more thorough evaluation. Evaluations of explanation faithfulness need to be performed and reported. This may be added as a supplementary table or figure.

We are thankful for the recommendation. In the revised version, we have added many more samples and more cancer types in our analyses. Briefly, we have added 450 TCGA samples, using just imaging information from these samples to perform prediction and comparing the results with bulk RNA-seq data corresponding to each of the 450 H&E images (new Figure 5 and Supplementary Figure S5). We have also added detailed pathological annotation for five samples from the TCGA database and performed intensive analyses. All of these new analyses suggest that the conclusion remains consistent.

As mentioned above, we have added a lot more interpretability analysis with new Figure 5, and Supplementary Figures S5, S9 and S10. This interpretability assessment, however, remains qualitative as in most other fields. We pioneered this analysis in the spatial transcriptomics field and we highlighted the importance for further research and added discussion about the current limitations of this approach.

We have also edited the sentence in the caption of Figure 1g.

3) Since no statistical tests were performed, it is difficult to assess the reliability of the results. Where appropriate, statistical tests need to be conducted and their p-values should be reported.

We added statistical tests across the samples throughout the manuscript. These include Moran's I and survival analysis, AUC, IoU and testing of significant correlations. We have added the descriptions for these tests in the method session. For example, statistical test results are shown in Figure 5.

4) Fig 2f is hard to read. It should be split into two rows for better readability.

Thank you for your comment. We have revised this and have split the panel into multiple rows, with clear sample IDs and moved the figure to the Supplementary Figure S1.

5) The text could be worded better and contains many spelling and grammar errors, making it difficult to read. Should be put through a grammar checker tool.

We apologize for this and we have revised throughout.

Recommendation:

The method appears to improve upon the currently existing methods in terms of predictive performance. It was also tested on different protocols, labs, and populations. Therefore, I recommend accepting after the concerns are resolved.

We thank the reviewer for the positive assessment and recommendation. We believe that our extensive revision with much more data analysis results have fully addressed the concerns and suggestions raised by the reviewer

Reviewer #3 (Remarks to the Author): Expert in breast cancer genomics, spatial transcriptomics, and digital pathology

In this study the authors present a new algorithm STimage which uses machine learning to predict gene expression on H&E images after learning the spatial gene expression patterns from spatial transcriptomics data. STimage and its performance are demonstrated on breast cancer datasets.

The concept is not novel and despite seemingly good performance of STimage, there are from my point of view many parts that are immature analytically which are accompanied with several typos and inconsistency through which make the manuscript difficult to assess.

We thank the reviewers for the comments, and we apologize for the writing issues that interfered with the interpretation and the reading flow of the manuscript. We have thoroughly checked the writing throughout.

We emphasize three key novel aspects of this manuscript, developed to advance the potential applications of digital spatial transcriptomics in cancer. First, we created probabilistic prediction models with uncertainty estimates, addressing the heterogeneity of cancer, including intra- and inter-tumor variations and technical discrepancies. We have expanded our testing to include many additional datasets to assess robustness. Second, we focused on interpretability measures, which are crucial for clinical applications of such models. Third, we highlighted the importance of selecting the most predictable genes for specific cancer types. Unlike existing methods that focus on variable genes, which are often not functionally relevant, our approach targets a subset of highly relevant and predictable genes specific to each cancer type.

Additionally, we introduced a new IoU metric for evaluating prediction performance in terms of geometric accuracy. We also included various assessment metrics beyond Pearson correlation

coefficient (PCC) and Moran's bivariate correlation, such as AUC, confusion matrices, and F scores. We believe that the IoU score provides the most accurate reflection of how well spatial patterns are captured.

For example, for the IoU scores, we have added the following text in the revised manuscript for new matrix to assess the predicted gene expression pattern (Figure 2d and Supplementary Figure S4):

...

We developed Intersection over Union (IoU) for gene expression prediction to address the inherent limitation of using PCC, which is known to be influenced by outlier noise and is unable to capture spatial patterns. This model performance metric reflects how well predictions match the ground truth with regard to the distribution of positive vs negative prediction after thresholding. Briefly, for a given gene, we standardize the prediction and ground truth values to Z-scores and then categorize them as binary values, based on whether they are greater than or less than zero. These binary values are then organized into a spatial matrix, creating a binary image where gene expression is plotted according to spot coordinates. To minimize noise and retain reliable prediction patterns, we apply Gaussian filtering with a sigma value of 1 to both rows (spatial X) and columns (spatial Y). After obtaining the refined prediction and ground truth images, we use the Intersection over Union (IoU) scores to assess the overlap between the spatial patterns of the ground truth and the predictions. This score effectively reflects the accuracy of predicting spatial patterns.

...

MAIN CONCERNS

1) The algorithm is tested only on a very limited breast cancer datasets of breast cancer. The authors should clearly state which data are used for training and testing throughout the manuscript to enable easier comparisons between different sections.

We thank the reviewer for the recommendation. We have significantly expanded the dataset and revised the manuscript to clarify the training and test sets and the genes used (Table S1). The revised dataset now includes: 46 breast cancer samples (comprising 2 fresh frozen and 2 FFPE samples from 10x, 6 from Wu et al., and 36 from Her2st), 6 kidney cancer samples, 10 melanoma samples, and 4 liver samples. We also included a Xenium breast cancer dataset with six patients and evaluated the model performance using this dataset. Overall, our work provides a comprehensive assessment of the application of deep learning models to predict gene expression.

2) There are different ST-seq dataset used across the manuscript Visium 10X, Visium-FFPE and Visium legacy. While the use of the larger dataset which: Visium legacy is mentioned in the abstract and material and method, it is not seen in the results. The authors should provide more information on how the different ST-seq datasets were used in the study and consider building a

training and testing dataset that includes a mix of the three methodologies to better evaluate the algorithm's performance on different types of data.

The Visium legacy was used in the manuscript multiple times (new Figure 3, Supplementary Figure S1, new Supplementary Figure S9), and we apologized for not clearly describing it. We have revised the text to clarify about datasets used.

We thank the reviewer for the suggestion about different combinations of data types in the training and testing datasets. We have added this comprehensive analysis and present the results in Supplementary Figures S9, S10.

Overall, we found similar performance between Visium and Visium legacy data.

Supplementary Figure S9.

Supplementary Figure S10.

3) Building on the previous comment in the material and methods the authors claim to use 1 FFPE tissue sample and when coming to the results they report testing 2 tissue FFPE samples. Such inconsistencies across the different sections in terms of how the datasets are used and reported highlight the immaturity of the manuscript.

We apologize for the inconsistency in the previous version. We used two FFPE samples from the 10x Visium platform, one of which was via the CytAssist protocol, while another one was from the FFPE protocol without the CytAssist. We have clarified these points.

4) The authors should provide a clearer description of the output of the algorithm, both in the abstract and throughout the manuscript, to ensure that readers understand what the algorithm does and what the results mean.

We thank the reviewer for the comment. We developed the STImage algorithm, which trains a neural network model on both H&E images and gene expression data measured by spatial transcriptomics technologies (Visium legacy, Vium, and Xenium) and subsequently predicts gene expression or cell types using only H&E images. This algorithm can infer gene expression profiles from routine H&E slides, significantly enhancing the utility of existing pathological archives by eliminating the need for direct spatial transcriptomics on every sample. This approach offers substantial cost and time savings, making sophisticated molecular analyses more accessible and routine. By bridging spatial transcriptomics with histopathology through H&E image analysis, STImage advances our ability to understand the relationship between morphology and molecular profiles of diseases, leveraging existing resources to provide deeper biomedical insights.

5) The definition of the 14 marker genes used in the study could be clarified. It is unclear how these genes were selected and whether they represent the top genes predicted by STImage. Additionally, the choice of these genes is not well-justified, and it is not clear why no stromal or endothelial markers were included. Providing more context on the selection criteria and the rationale behind these choices would help readers better understand the approach. Moreover, it could be useful to consider incorporating stromal and endothelial markers as these compartments are important in breast tumors and can also be annotated by pathologists. This would improve the algorithm's ability to predict gene expression in all compartments of breast tumors.

As mentioned in the responses to Reviewer 1, the 14 markers include those identified from the training dataset that are cancer related, and those well known as cancer markers from the literature. However, it is worth noticing that the selection of the 14 markers is not our focus of this analysis, but is used only to demonstrate the model can be trained and applied as a gene panel of markers, similar to the approach by current sequencing tests applied in the clinics. In the revised version, we expanded our model to predict 1522 functional genes, which are more informative than using a single marker, e.g., cancer marker, stromal, or endothelial markers.

6) The Swarbrick's lab dataset has been annotated in the original publication, could this be used for validation, interpretation here?

Yes, we used the annotation from Wu et al.'s original publication in our Figure S2e to show that our model predicted cancer marker gene expression shows a similar pattern compared to pathological annotation. We also performed interpretability on the test set from Wu et al.'s dataset.

7) The non spatial tile labelling and testing is not convincing, first it is not clearly explained how the training was performed. Second testing is performed only on two samples which are closely related since they are successive sections from the same tumor. One needs to really extend the testing to prove applicability.

We have added 450 TCGA samples, using just imaging information from these samples to perform prediction. We compared the predicted expression with bulk RNA-seq data corresponding to each of the 450 H&E images (new Figure 5 and Supplementary Figure S5). For deep annotation, we have added two more samples from the TCGA dataset. In total, three individual samples (five whole slide images) are included for testing. We then compared the spatial distribution of the predicted values with the cancer vs non-cancer annotation by the pathologist. Overall, we found consistent patterns across all samples and slides (Figure S5).

8) The preprocessing, processing use of the ST-seq data is not described.
We have added this in response to the comment number 2 of this reviewer.

9) It is unclear what is the impact of OpenCV2 on tiles removal, especially the tiles situated in the edges that may contain less than 70% tissue coverage, it seems to me that some tiles in the 'middle' of the tissue with bad coverage may be removed, but also tiles on the edges or tiles containing adipose or less dense tissue? How many tiles are removed per image and where? Are important tiles removed? Is it necessary to remove them? Are these tiles creating so much noise that they need to be removed?

We have removed a small number of tiles, as detailed in the table added to the revised version (displayed below). Since each tissue typically results in over 2,000 tiles, the number of removed tiles is generally less than 1%. The likelihood of these removed tiles being located in biologically important regions, such as edges, is even lower. In contrast, including 30% of empty tissue regions would significantly impact overall performance, as the network would need substantially more samples to learn how to predict tiles missing tissue, given that these tiles constitute less than 1% of the data.

Figure. Examples of bad tiles having (<70% tissue coverage) that were removed and good tiles (>70% tissue coverage) that were retained.

Table 1. Overview of the different datasets and the spots/image tiles retained after filtering.

No.	Dataset	Samples	Tissues	Resolution	Unfiltered spots	Remaining spots	Genes
1	Her2ST [28]	36	Frozen	100 μM	13,653	12,584	11,871
2	Swarbrick's Lab [14]	6	Frozen	55 μM	15,611	14,951	14,664
3	Public BC 10X-Visium	2	Frozen	55 μM	7,785	7,289	36,601
4	Public BC 10X-Visium	1	FFPE	55 μM	2,518	2,338	36,601
5	Melanoma	13 (5 slides)	FFPE	55 μM	15,209	14,405	17,943
6	Liver Visium [16]	4	Frozen	55 μM	19,968	19,967	36,601
7	KC Visium [15]	6	Frozen	55 μM	16,131	15,632	36,601
8	BC Xenium	5	Frozen	subcellular	-	21,076 (1,294,600 cells)	280
9	SC CODEX	3 (1 slide)	Frozen	subcellular	-	6,174 (813,004 cells)	34 (Proteins)

Table 1. Overview of Datasets used by STimage.

BC - Breast Cancer, KC - Kidney Cancer, SC - Skin Cancer

10) Stain normalization seems to be performed to look like the RGB intensity of a template image. What is the influence of the choice of the template image. How was the template image chosen?

Stain normalization is crucial for enhancing the accuracy and robustness of deep learning models. Variations in stain intensities and hues often arise from differences in staining protocols, chemicals used, and scanner settings across different laboratories and imaging microscopes. Stain normalization aims to minimize these variations, ensuring consistency across datasets from various sources (Supplementary Figure S3). By normalizing stains, models can focus on learning features that are consistent across different datasets, thereby improving generalization capability. The template images used for normalization were selected randomly and should not influence the model's performance.

11) The uncertainties are assessed spatially. Have the authors assessed if the uncertainties seem to be mainly driven by spatial location or the genes assessed? In other words, what is the highest source of uncertainty, the quality of the image? Or the capacity of the algorithm to predict a specific gene expression?

We thank the reviewer for the interesting questions. We have assessed the source of uncertainties and data clearly suggest that the measured expression values of the genes lead to high or low variance in the predicted values (Figure 2a). The Zoom-in images show that the yellow-red spots in the Figure 2a below remain high (while blue spots are consistently low) for aleatoric, epistemic and predictive values.

Figure 2a.

12) Breast cancer is a very heterogeneous disease, when assessing only 14 genes across the manuscript it may be that the performances are different from subtype to subtype. No track of the clinicopathological features of the sample used is given or hint towards the model being associated with subtypes is given. Some have shown that subtype can be predicted from H&E images, it is therefore important to assess how this can affect the model.

We have assessed how different cancer types have distinct sets of the most predictable genes, each enriched for relevant pathways specific to the cancer type. Supplementary Figure S1 shows the most predictable genes for HER2-positive breast cancer, Supplementary Figure S6 displays the top predictable genes for skin cancer, and Supplementary Figure S8 illustrates the most predictable genes for kidney cancer.

13) The interpretability analysis in the end of the 'prediction performance' paragraph is disappointing, the only feature which the author mention in their interpretability section is nuclei which has been reported before, the author attempt to conclude on other feature in a gene-specific manner... perhaps... but I am not managing to understand the last sentence which involve COX6C and KRT5 genes. It would be interesting to understand further which feature drive the prediction of different genes

As discussed above (Reviewer 2 comment 1), in the revised version, we explored multiple aspects of explainability including saliency mapping with Grad-CAM (Supplementary Figure S13), feature attribution with LIME (Supplementary Figure S13), comparing with pathological annotation (Figure 1g,h; Supplementary Figures S2, S5), prediction of functionally relevant genes (Supplementary Figures S6, S7, S8), and analyzing the latent space (Figure S12f).

14) Can uncertainties by training on different ST-seq platform/ methodologies be assessed?

We have assessed the performance of using models trained on Visium and tested on images of Xenium and CODEX data and non-spatial TCGA data. The results are presented in the response to Reviewer's comment 2 above. The results are presented in Figure 3, Supplementary Figures S5, S9, S10, S11.

15) What is the scalability of STImage, can more genes be assessed? It is important to provide the performance of STImage for all genes tested at least the 14 mentioned all along ideally all genes.

STImage is scalable. In the revision, we have assessed STImage in the scenarios of simultaneously training for 1000 Highly Variable Genes and 1522 functional genes across more samples and disease types.

16) The interpretability, last chapter, is not clear, suddenly in this chapter a new gene for immune marker appear C3 is it a typo? do the author mean CD63? or the C3 gene is used, C3 is not a typically marker gene for immune cells.

We used C3 in the previous version as we were applying a method to select contrasting gene pairs that had opposite spatial distribution. In this revision, we focused on more functionally relevant genes, and so we did not use C3, but used CD52 as an immune gene for interpretability analysis.

MINOR comments

- No p values are shown for the PCC (pearson correlation):

We have added p-values for PCC (Figure 5)

- Typo: for independent validation (p2)

Corrected, thanks

- Typo: PPFE sample (p2)

Corrected, thanks

- Typo: A classification model can be added to produce complementary information (p8)

Corrected, thanks

Reviewers' comments:

Reviewer #1 (Remarks to the Author):

Although the author has done more analysis on the original version, the performance of the model needs to be further improved. Some similar works in the past two years seem to have better performance than STImage, such as THItGene and IGI_DL. The following are the additional comments for the author's modification:

We have compared STImage with THItGene and IGI_DL. In both cases, STImage shows superior performance (Figure R1). Notably, in this revised version, we have markedly improved the STImage performance by integrating state-of-the-art foundation models into STImage. Foundation Models, such as Virchow2, were trained on 3.2 million whole slide H&E images, with 632 million parameters, and therefore are more suitable image extractors than traditional methods, such as ResNet50. Our advanced Virchow2_STImage model outperformed all other models. Further, we created an ensemble model named FSTImage_ens, where we integrated multiple foundation models into the STImage base model. Each foundation STImage model contains one foundation model as the upstream image extractor and an STImage negative binomial module. The foundation models used are: GigaPath, H-Optimus, Phikon, cTranspath, CONCH, UNI, and Virchow2. We then aggregated by averaging the results from these models to achieve a final FSTImage ensemble (FSTImage_ens) prediction. We found STImage, Virchow2_STImage, and FSTImage_ens performed better than any other models that we compared to, including THItGene and IGI_DL. We introduce the FSTImage in our revised Figure 1b.

Our new results are shown in the figure below (and have been added to the main Figure 2e in the revised version).

Figure R1. Benchmarking STImage with IGI-DL, THItGene and other models. Introducing two new models Virchow2_STImage and FSTImage_ens.

1. The process of selecting the variable genes and 1,522 functional genes is unclear. Please provide a clear explanation of how these genes were chosen, including the specific methods and thresholds utilized. The rationale for selecting the highly variable genes (HVGs) and the genes used for prognosis prediction should be clearly articulated.

We consider a set of all 1630 functional genes (1,522 remained after removing lowly expressed genes) combined from panel gene lists curated by main spatial transcriptomics platforms available for Xenium and CosMX. These genes are the union of 1253 genes from the human pan-cancer panel and 1056 genes from the human immunology panel (10xGenomics). The pan-cancer panel contains: 1) 21 cancer-related pathways such as Hedgehog/Ras/p53/MAPK/Notch/mTOR/TNF/TGF-beta/TLR/Wnt signalling pathways; 2) markers for 28 cancer cell/tissue types, such as colon cancer, breast cancer; 3) 17 cancer-related cellular processes such as immune response, chemotaxis; and 4) 37 functional annotation (e.g., nucleus, extracellular matrix) and processes (e.g., angiogenesis, mitosis). The immunology panel contains: 1) 20 immune pathways, 2) markers for 29 cell/tissue types (e.g., liver, lung, NK cells, macrophage, DC), 3) 41 immune-related cellular processes (e.g., inflammatory responses, innate immune responses), 4) 25 immune-related functional processes. Although these lists are not exhaustive, the genes represent an

unbiased selection of functionally relevant genes, not just focusing on 1000 highly variable genes as in previous reports.

2. The rationale for selecting only 450 samples for the prognosis prediction analysis, rather than using the full TCGA cohort, is unclear. If any sample selection was performed, the criteria and justification must be provided. Please present the results for all TCGA samples.

We initially performed random sampling to analyse a subset of the TCGA dataset. To address the reviewer's request, we downloaded all 3,066 images for TCGA breast cancer and applied the pre-trained STimage model to all samples. The conclusion from this analysis remains the same compared to the initial random subset analysis (also refer to responses to comment 3, and Figure 4b).

H&E images that didn't have matched bulk RNAseq data or clinical data were excluded. We used HistoQC to remove low quality images from the analysis, removing 215 images using filtering criteria below:

- blurry_removed_percent \geq 0.9
- small_tissue_removed_num_regions > 97.5th percentile
- pixels_to_use < 5th percentile

All images that passed the quality filtering criteria were used for further analysis. The results show that our predicted data can significantly stratify overall survival across the three subtypes of breast cancer (Figure R2).

The sample-wise correlation was performed for the samples that had matched bulk RNA-seq gene expression and H&E tissue. The gene-wise correlation was performed for the samples with matched bulk RNA-seq expression data, H&E tissue and breast cancer subtype information (three breast cancer subtypes, Her2, Lum and TNBC), (Figure R2). Survival analysis was performed for the samples with matched H&E, bulk RNA-seq gene expression, and survival data.

Figure R2. Kaplan Meier survival curves for all patients in each subtype. The first row is the results using the measured bulk RNAseq data and the second row is for the predicted data from just H&E (at spot levels and then averaged across all spots to get one value per gene per image). The genes used for plotting these curves are top 5 genes with the highest c-index from the univariate cox-regression analysis.

3. The correlation analysis presented was conducted on a per-sample basis. Please provide the results of the correlation between the predicted gene values and the true bulk gene expression values for each gene. Are the genes with a high correlation to bulk values the ones that your gene prediction model predicts with greater accuracy?

We performed analysis on a per-sample basis for the whole transcriptome rather than per gene across samples because, in the TCGA dataset, the bulk sequencing data and the image data for a patient often come from different tissue blocks of the same patient. The two tissue blocks from a breast cancer patient can be markedly different and so the information encoded by the bulk sequencing and the image data could be relatively large. Given a gene, changes in the expression value from one patient to another patient was measured by RNAseq data, but not by imaging data. The deviation of the correlation value between RNAseq data and image-based prediction is therefore the composite of three sources of variation: between patients, between imaging sections and tissue blocks used for sequencing, and true difference between predicted values of the genes from the imaging sections and the sequencing results of the same imaging section (which does not exist). Therefore, the correlation for one gene across samples is affected by multiple factors not just how accurately the model predicts the gene expression in an image, but also affected by the variation between patients in terms of image quality, the technical sequencing variation across samples, and biological variation between tissues used for imaging and different tissue blocks used for sequencing. Therefore, the sample-wise correlation becomes a metric that is not possible to be decomposed to assess the model performance. Even the gene-wise correlation for each sample is less ideal, but is a better metric to use.

We also re-run the gene-wise correlation analysis on a per-sample basis for all TCGA samples and the results are shown in the **FigR4** (added to the revised version as Figure 5). The average correlation for the new results is similar to previous results i.e., 0.48.

Figure R4. Sample-wise and Gene-wise Pearson correlation analysis using the TCGA breast cancer dataset. This figure shows the gene-wise correlation for each sample and across 1034 samples. The number on the top right indicates the number of samples (each dot in the plot represents a sample).

4. The presence of outliers in the data may affect the accuracy of the Pearson correlation coefficient. The Spearman correlation coefficient results require further explanation to interpret result significance in the context of your study.

We agree with the reviewer and therefore we added more assessment metrics, including IoU as discussed in more details below.

5. Moran's I is used to analyzing the spatial distribution pattern of a single variable. However, the author uses it to assess the accuracy of gene predictions without comparing it to the Moran index of actual gene values, which is confusing.

Here, Moran's I was used to assess how the spatial autocorrelation in the predicted values correlates with that of the original spatial transcriptomics data. A gene with a higher Moran's I value suggests that the spatial pattern is better preserved than a gene with a lower Moran's I. This is an additional metric to PCC, a commonly used metric that does not account for the spatial context.

The interpretation of spatial autocorrelation is as follows: if the score is high, it means that the prediction matches the ground truth and that the prediction of the gene across the spots also preserve the spatial pattern of that gene between neighbor spots. If the score is low, it could mean either that the prediction is inaccurate or that the gene does not have a spatial pattern.

6. The details of the prognosis prediction model are missing, particularly the method used to derive the survival score from the predicted gene expression values. More thorough explanation is needed.

We have performed more thorough analysis as described in our updated text of the Methods section as below:

To perform external validation, assess the robustness and demonstrate clinical application of STImage, we applied the model on the TCGA dataset. For this analysis, H&E images were downloaded from TCIA and bulk gene expression and clinical data for corresponding patients were downloaded from "<https://www.cancer.gov/ccg/research/genome-sequencing/tcga>". We downloaded 3,066 images. 1,532 images that didn't have matched bulk and clinical data were excluded from the analysis and remaining 1,534 images were used for quality check using HistoQC. For 285 images HistoQC results were NA or not generated as the images were corrupted and couldn't be read. For the remaining 1,249 images with HistoQC results, we filtered out low quality images using the following parameters: `blurry_removed_percent` \geq 0.9, `small_tissue_removed_num_regions` > 97.5th percentile and `pixels_to_use` < 5th percentile. These thresholds were chosen based on the empirical distribution of quality parameters, removing extreme values. Using these filters, 215 images were excluded. Only images that passed all the filtering criteria were then used for further analysis. We finally used 1,034 breast cancer images from 670 patients and predicted the gene expression of 1,522 highly variable genes.

Predicted gene expression for each image was averaged across all spots to compare with the ground truth TCGA bulk gene expression. We measured the performance of STImage by calculating Pearson Correlation (PCC) between the average predicted expression across all the spots with the true bulk gene expression for the matched samples. We also computed the performance of STImage for predicting top-300 predictable genes for three subtypes (HER2+, Luminal and TNBC). Patient subtypes were defined according to the criteria described in Xu et. al. (2022).

STImage prediction was then used to perform survival analysis and stratify the patients into low- or high-risk groups. We first built a univariate cox-regression model for the top 300 predictable genes using the function 'coxph' from R package "*survival*". Using the top-300 predictable genes, we built the univariate Cox regression model for each gene individually. Cox regression models were built separately for true and predicted gene expression. Days to death and days to last follow-up (for patients still alive) were used as survival times, and survival status (alive/dead) was used as the event indicator. The univariate models were evaluated using the concordance index (c-index) for both predicted and true survival outcomes. The top-5 genes that had the highest c-index were then used to build the multivariate cox-regression model. We used a 3-fold cross validation strategy with 100 repetitions to evaluate the multivariate models. The median survival risk score from the multivariate model was used to stratify patients into low- or high-risk groups. Univariate and multivariate models were built separately for each subtype. For the bulk and the predicted data, the average c-index was found to be 0.75 for all subtypes. We used the function 'concordance' from the R package "*survival*" to assess the concordance performance. Univariate Cox regression models and Kaplan-Meier curves were computed using R package "*survival*". The multivariate regression models were built using the R package "*ClassifyR*".

7. The poorer performance of the STImage model in some cases of the Her2ST LOOCV benchmark is concerning. Whether STImage is biased to the spatial pattern of a specific slice?

STimage performed better than other methods in the legacy dataset (Her2ST data) in most cases. We found that the performance was only equal or slightly lower in the samples that the quality of the gene expression matrices was sparse (i.e., too many 0s values across the whole matrix). Notably, for the sparse data, the spatial feature appears to be less reliable, even with the ground truth data, suggesting that the comparisons for those samples with low quality should be interpreted with caution, or should even be excluded. We found that the STimage performance is consistently better for less sparse samples. Importantly, our new FSTimage_ens performed better than all other models in all cases, including the ones with over-sparse data, although we suggest that for sparse samples, they should not be used for assessing models.

Figure R5. The PCC for all models on the Her2ST dataset in LOOCV settings. The PCC for each test sample is grouped and ordered by the sparsity of the real measurement of ST data. The sparsity is calculated by the number of zeros in the ST gene expression data, ordered from high sparsity (left) to low sparsity (right). Refer to the Figure R1 above for a clearer presentation of the Figure (not ordered by sparsity).

8. While multiple models were trained using different datasets, the benchmark analysis was limited to the Her2ST dataset and 785 genes instead of 1522 genes that used in prediction. This raises questions about the ability to accurately reflect the models' predictive performance and generalization capabilities.

We appreciate the reviewer's comment regarding the benchmarking analysis. To address this, we have added additional benchmark experiments across 10 samples from 5 Visium slides covering other cancer types, including skin melanoma datasets, to further evaluate the predictive performance and generalization capabilities of our model. The 785 genes mentioned by the reviewer were from our test for the top 1000 highly variable genes (HVG), not for the 1522 functional genes. The reason for using 758 genes, but not 1000 HCG, was that we removed the genes that expressed in less than 10% of the total spots, an unbiased way to remove noise. In addition, we introduced the analysis of 1522 functional genes. The results demonstrate that STimage consistently outperformed the models suggested by the reviewer, including THtoGene and LGL_DL (Figure R6). Moreover, our new model FSTimage_ens more markedly outperformed all other models (Refer to Figure R1).

For the generalization, we assessed the models for different cancer types, including skin cancer, kidney cancer, and a non-cancer liver dataset and across different experimental protocols, patient cohorts and data modalities (Table 1 below).

Figure R6. PCC for benchmarking STImage, THltoGene, and IGI_DL on 10 melanoma samples across 10 Visium slides. Each group of boxes shows the PCC for the top 300 predictable genes for each model across test Visium slides.

Table 1. Datasets used for evaluation of STImage performance.

No.	Dataset	Samples	Tissues	Resolution	Unfiltered spots	Remaining spots	Genes
1	Her2ST [30]	36	Frozen	100 μM	13,653	12,584	11,871
2	Swarbrick's Lab [14]	6	Frozen	55 μM	15,611	14,951	14,664
3	Public BC 10X-Visium	2	Frozen	55 μM	7,785	7,289	36,601
4	Public BC 10X-Visium	1	FFPE	55 μM	2,518	2,338	36,601
5	Melanoma	13 (5 slides)	FFPE	55 μM	15,209	14,405	17,943
6	Liver Visium [16]	4	Frozen	55 μM	19,968	19,967	36,601
7	KC Visium [15]	6	Frozen	55 μM	16,131	15,632	36,601
8	BC Xenium	5	Frozen	subcellular	-	21,076 (1,294,600 cells)	280
9	SC CODEX	3 (1 slide)	Frozen	subcellular	-	6,174 (813,004 cells)	34 (Proteins)

Table 1. Overview of Datasets used by STImage.

BC - Breast Cancer, KC - Kidney Cancer, SC - Skin Cancer

9. The authors only benchmarked HisTogene, STnet, and Hist2ST released in 2021 or 2022. How do the more recently released THltoGene and IGI_DL perform?

As discussed above in the Comments #7 and #8, and the general comment with Figure R1, we have added new benchmarking analyses on HER2ST and the Lagrange ST dataset for breast cancer, as well as Visium data for skin cancer. We consistently found higher performance for STImage baseline across datasets. Moreover, our FSTImage_ens markedly improves the performance of the STImage baseline model.

10. The median PCC of BRCA FFPE ST data predicted by Stimage is less than 0.1 in Fig. 2E, this suggested a very weak correlation.

The low performance for this particular setting was expected, as we trained the model on a small Visium fresh-frozen dataset (2 samples) and tested it on the FFPE dataset (1 sample). This experiment evaluated

the model's performance on an out-of-distribution (OOD) test, as the image and gene expression data quality differed significantly between the fresh-frozen and FFPE datasets. Therefore, given this stringent setting to test both OOD and the small-size training datasets, a weak correlation is expected. In most applications, the training dataset is often more than 10 slides, a sample size that consistently resulted in predictable genes with PCC ranges from 0.25 to 0.8 (Figure 2c, e). Moreover, PCC is not the accurate metric to reflect model performance regarding how well the spatial pattern can be predicted. In this case, intersection of union (IoU) is a more relevant metric. We show that we can achieve IoU consistently higher than 0.7 (Figure 2d and Figure S4b).

11. In Figure S8, the PCCs around different slides show quite different results, with some slides showing median PCCs below 0.

Please refer to our responses to comment #7. The low performance for these samples was due to poor data quality in some samples (sparse data with too many 0 values). In Figure S8a and S8b, we show that even with a very low PCC value, the prediction of cancer markers is still consistent with the pathological annotation and, in some cases, even better than the ground truth spatial transcriptomics measurements, possibly due to the model also learning from imaging data not just from the sparse sequencing data.

12. IoU values in Fig.S11g are abnormally high. From the figure, it seems there aren't many overlapping areas between prediction and ground truth.

We thank the reviewer for identifying this issue and we apologise for this unintentional mistake. The high IoU scores for classification model were due to applying the same IoU function from the regression model to the binary classification model. The parameter 'average' in the function "jaccard_score" from sklearn module was set to "binary" for the classification model rather than "None" for the regression model, where the predicted values are continuous. We have now fixed that issue. The correction has been made for the figure S11 and the IoU remains relatively high (0.3 to 0.6).

Figure R7. Recomputed IoU scores for the classification model. The Jaccard scores were obtained for both "High" and "Low" class for a gene (high and low refer to the binary classes of the categorised gene expression). The average of scores of both classes were computed as a final IoU score of a gene. An example of the gene CD52 for the two test-set samples is shown. We have corrected the IoU scores in the Figure S11 panels C and G.

13. The section on Xenium cell type classification does not present the results of the leave-out test comprehensively. It lacks comparative results with other classification models and evaluation metrics, such as mIoU.

We performed a leave-out test using five Xenium datasets, and the results are presented in Figure R8 (we added this Figure as the Figure S14 in the revised version). The performance remained consistent across all test samples, demonstrating the robustness of our classification model. Among the different cell types, cancer epithelial cells were the most accurately predicted (82.0%-92.6% accuracy). The second most predictable group was immune cells, which included B cells, T cells, and myeloid cells (58.6%-83.1%). Ability to predict cancer cells and immune cells from H&E images enables the prediction of cancer-immune interactions within the tumour microenvironments, suggesting diagnosis and prognosis values. Given that much more data will be available in the future, our model has the potential to achieve even better predictive performance, particularly for rarer cell types.

We also calculated the mean IoU score for each LOOCV test sample (Figure R9; added as Figure S15 in the revised version). In benchmarking against the newly published CellViT (Hörst et al. 2024) model, which is a Vision Transformer-based model that can also predict single-cell segmentation and cell type classification, our performance in terms of IoU score is 2 to 4 times more accurate than CellViT (Figure R10; added as Figure S15 in the revised version). Notably, our classification model utilises Xenium data from a small set of samples, different to CellViT that utilised hundreds of thousands of images. With our approach, the training data is generated automatically using an image registration method to align the DAPI cell segmentation mask (which is more accurate than HE) with cell types derived from marker gene expression (experimental measurement) from Xenium data. This approach reduces the need for manual labelling and minimises human error.

Confusion Matrix (QMDL01)

True \ Predicted	Immune	Stromal	Cancer Epithelial	Endothelial
Immune	95731 (70.2%)	16379 (12.0%)	16451 (12.1%)	7844 (5.7%)
Stromal	32558 (30.5%)	45280 (42.4%)	15479 (14.5%)	13448 (12.6%)
Cancer Epithelial	14327 (9.4%)	8471 (5.6%)	125081 (82.0%)	4675 (3.1%)
Endothelial	4066 (13.4%)	3254 (10.7%)	5447 (18.0%)	17500 (57.8%)

Confusion Matrix (QMDL02)

True \ Predicted	Immune	Stromal	Cancer Epithelial	Endothelial
Immune	19200 (85.1%)	1093 (4.7%)	1961 (8.5%)	845 (3.7%)
Stromal	3840 (32.8%)	5827 (49.8%)	1173 (10.0%)	860 (7.3%)
Cancer Epithelial	3054 (5.1%)	1198 (2.0%)	54843 (92.1%)	451 (0.8%)
Endothelial	1244 (17.1%)	1190 (16.4%)	694 (9.6%)	4130 (56.9%)

Confusion Matrix (QMDL03)

True \ Predicted	Immune	Stromal	Cancer Epithelial	Endothelial
Immune	73576 (72.5%)	16052 (15.8%)	9427 (9.3%)	2418 (2.4%)
Stromal	12015 (22.7%)	34597 (65.4%)	3808 (7.2%)	2453 (4.6%)
Cancer Epithelial	8066 (5.1%)	8172 (5.2%)	138880 (88.4%)	2013 (1.3%)
Endothelial	2678 (11.1%)	4731 (19.6%)	3067 (12.7%)	13597 (55.5%)

Confusion Matrix (QMDL04)

True \ Predicted	Immune	Stromal	Cancer Epithelial	Endothelial
Immune	1179 (58.6%)	433 (21.5%)	219 (10.9%)	180 (8.9%)
Stromal	913 (12.8%)	4473 (62.8%)	1053 (14.8%)	678 (9.5%)
Cancer Epithelial	1289 (2.7%)	1633 (3.4%)	44971 (97.6%)	664 (1.4%)
Endothelial	459 (7.9%)	994 (17.1%)	505 (8.7%)	3856 (66.3%)

Confusion Matrix (QMDL05)

True \ Predicted	Immune	Stromal	Cancer Epithelial	Endothelial
Immune	21005 (60.3%)	9097 (26.1%)	3892 (11.2%)	831 (2.4%)
Stromal	5752 (17.9%)	20854 (65.0%)	4121 (12.8%)	1345 (4.2%)
Cancer Epithelial	9029 (6.9%)	7644 (5.9%)	111695 (85.9%)	1590 (1.2%)
Endothelial	1714 (11.9%)	3504 (24.3%)	1571 (10.9%)	7644 (53.0%)

■ Immune
 ■ Stromal
 ■ Cancer Epithelial
 ■ Endothelial

Figure R8. Leave-one-out cross-validation for single-cell segmentation and cell type classification in the breast cancer Xenium dataset. Each row represents one test sample in each cross-validation experiment. The first column shows the predicted cell type, the next column contains the ground truth cell type labels from the Xenium data, and the next row presents the HE image, which is the only data required in the prediction step. The last column shows the confusion matrix for all five cell types. The numbers indicate the number of cells, and the percentages represent the proportion of each prediction relative to all ground truth values.

Figure R9. Mean IoU score for each sample in the LOOCV experiment. The blue bar represents CellViT, while the orange bar represents STImage Classification.

Figure R10. Randomly selected predictions at the tile level to compare the performance of the STimage classification model, CellViT. Each tile has three columns: from left to right, the STimage classification model, CellViT, and the ground truth.

14. Is there a cell type marker in the genes predicted by the model? This can be compared with the classification results.

We thank the reviewer for their comment. We performed cell type prediction on one of the held-out Xenium test samples to demonstrate that the model can accurately identify cell types (Figure R11a, b). Using the same H&E image, we applied the STImage gene expression prediction model that was trained on the low-resolution Her2st dataset to predict marker expression in the Xenium sample (Figure R11d). CD79A, an immune cell marker, and ESR1, a breast cancer marker, were selected for visualisation. The predicted expression patterns of CD79A and ESR1 closely align with the corresponding immune and cancer cell types predicted by the classification model. This consistency across models suggests that the predictions are biologically meaningful. Importantly, the classification and regression models were trained on different datasets, yet both produced coherent and spatially consistent results in for same HE image. This highlights the robustness of the models and indicates their ability to generalise and reflect real biological signals.

Figure R11. Comparison of cell type prediction from the classification model at the single-cell level and marker gene prediction from the regression model at the spot level. (a) Predicted cell types for one leave-out test Xenium sample. (b) Ground truth labels annotated from Xenium gene expression data. (c) H&E

image for the same slide used as input for prediction. (d) Predicted markers for CD79A and ESR1 at the spot level using the model trained on the Her2st dataset. (e) Pathological annotation for the same slide.

15. The authors didn't assess prognosis performance for STImage, such as C-index.

We thank the reviewer for the suggestion to assess the prognostic performance of STImage. We have calculated the c-index for true and predicted data for both univariate and multivariate models. We performed the survival analysis for each subtype separately. The top-5 genes for each subtype resulted in a better performance following the multivariate analysis; giving consistent results with a c-index of 0.75 (Figure R12; added to the revised version as Figure 5d).

Univariate Analysis: Using the top-300 predictable genes, we run the univariate cox-regression model for each gene individually.

Multivariate Analysis: Using the top-5 genes with highest c-index obtained from univariate analysis, we built a multivariate cox-regression model using the function CrossValidate from ClassifyR to predict the risk/survival score.

Figure R12. We assessed the prognostic performance of the STImage predictions for TCGA data as well as the bulk data for the three subtypes (Her2, Lum and TNBC). Each dot in the univariate analysis box-plot represents the c-index of a gene. For the multivariate analysis, we used a 3-fold cross-validation strategy with 100 repetitions to evaluate the performance. Each dot in the multivariate analysis is a c-index of the test set. The red dotted line indicates the mark of 0.5 c-index.

Reviewer #3 (Remarks to the Author):

Thank you to the authors for addressing the questions and comments previously raised. Although I appreciate the technical efforts to improve robustness, reproducibility, and interpretability, the manuscript remains highly technical without demonstrating the biological application of the developed tool suite. The tool suite's applicability appears complex and not easily accessible to researchers. The main concerns are the lack of easy access and the absence of user-friendly software, as well as insufficient biological interpretations, which limit the study's interest.

We thank the reviewer for the comments. For user friendly instructions to use the software, we have provided detailed documentation from installation to running the model. All the code and data are open source (<https://github.com/BiomedicalMachineLearning/STImage> and data at <https://espace.library.uq.edu.au/view/UQ:4fb74a9>). We also provided interactive webtool for user to explore the model and interpretability without the need to write code to understand how the model works (<https://gml-stimage-web-app.streamlit.app/>). The interactive tool allows the users to upload their own H&E images to make the prediction and run through the analysis.

Our documentation can help users install the software easily. Once the software has been installed, users can run sequentially from five commands as convenient wrappers to preprocess data, train model, run prediction, perform model interpretability analysis, and produce visualization for the outputs. We also provide a configuration file for users to conveniently specify customised and reproducible setting to train the model.

Regarding interpretability, to our knowledge, STImage is the only software that has a built-in function for implementing interpretability techniques (e.g., segmentation followed by LIME interpretation for feature importance of every single cell). Examples of the interpretability analysis are shown in our Figures 4 c,f and Figures S11 and S12.

While the authors highlighted key novel aspects and technical improvements in their response, these improvements should be benchmarked against other algorithms.

As described in the responses to Reviewer 1 comments #8 and #13 and as shown in Figure 2, Figure S1, and Figure S15, we have thoroughly compared our models with other algorithms. Our base model STImage surpassed all models and our improved STImage_ens produces a much higher performance than the base model and all other models.

More critically, there are no new biological findings that enhance our understanding of diseases or physiological processes, which raises serious questions about the usefulness of the presented tool suite and algorithm.

We appreciate that the Reviewer 3 evaluated our work from a biological and clinical application standpoint. We would like to kindly mention that our model focuses on the machine learning application in the context of digital pathology, where the main purpose is not to discover new biology, but to predict

gene markers and cell type markers that can be used for diagnosis and prognosis. Nevertheless, we have also explored the ability of STImage in enhancing biological understanding. by finding robust spatial features for prediction as in the Figure R13, as well as finding gene markers that are functionally relevant to cancer and can be reliably predicted across samples. However, this is not our main focus of the paper.

In the response, the authors mentioned inter and intra-tumor heterogeneity of cancer. However, this aspect is critically missing in the manuscript. There are no measurements of intra-tumor heterogeneity or its association with gene expression and/or clinically relevant parameters. The algorithm's ability to detect phenotypically different tumor clones is neither assessed nor validated. Intra-tumor heterogeneity could be compared to matched WGS for cancer cell heterogeneity or scRNA-seq data for immune cell heterogeneity. Additionally, with the few samples analyzed for each disease, it is probably not relevant to discuss inter-tumor heterogeneity.

In our manuscript, we did not make a claim about inter-tumour heterogeneity. We agree with the reviewer that the variability between patients is a parameter that requires assessing more samples, which will be available in the future when the cost to run a spatial experiment will be lower and more researchers will have access to the technologies. In this study, to our knowledge, we have evaluated the most diverse collection of spatial transcriptomics datasets, covering different cancer types, sample cohorts, technological platforms, taking into account samples in out-of-distribution settings (Table 1). The intra-tumour heterogeneity was thoroughly assessed via clustering analysis and the spatial variation in gene expression across different tissue regions (Figures S8, S14).

The authors also mention "highly relevant predictable genes for each cancer type," but it is unclear in the manuscript what these genes predict or how they reveal relevant biological insights.

Please refer to our detailed responses to Reviewer 1, Comment #1, regarding predictable genes. We would like to highlight that the focus of the model is to predict gene expression and cell types from H&E images as known markers for cancer diagnosis in digital pathology rather than focusing on discovering new biological insights. For in-depth biological analysis, we would use directly spatial transcriptomics data rather than predicted signals from H&E images, as demonstrated in our other projects, like our stLearn software that implements different types of analysis such as differentially expressed genes between two conditions or cell-cell interactions but not based on H&E images.

In short, while I appreciate the technical improvements and the additional data used in the revised version, the manuscript still lacks critical elements such as cell type specificity, clinically relevant analysis, spatial ecosystem generalization, and specific spatial gene expression. These missing aspects result in a lack of novel biological insights, which limits the overall interest and impact of the study.

Regarding cell type specificity, as discussed in the responses to Reviewer 1 comment #13, among the different cell types, cancer epithelial cells were the most accurately predicted (Figures R8 and R10). The second most predictable group was immune cells, which included B cells, T cells, and myeloid cells. This has potential for predicting cancer-immune cell interactions in tumour microenvironments, providing high-resolution information important for cancer diagnosis and prognosis. Given that much more data will

be available in the future, our model has the potential to achieve even better predictive performance, particularly for rarer cell types.

Regarding spatial ecosystem and specific spatial gene expression, throughout the manuscript, we show the prediction of biologically relevant genes that are markers for cancer cell types. These are listed below: Figure 1 f, g, h showing COX6C; Figure 2 a, b showing CD63, ATP1A1; Figure 3 showing ESR1 and S100B; Figure 4 displaying prediction of immune, stromal, epithelial and endothelial cells; Figure 5 predicting VEGFA and MMPED1 in three breast subtypes HER2, Luminal and TNBC. Similarly, Supplementary Figures also present pathways and predicted genes for skin cancer (Figure S5; showing prediction of B2M, CD74, PFN1, and enrichment of skin macrophage, antigen presentation and signalling pathways), for liver diseases (Figure S6, showing APOE, C1R and SERPING1 prediction, with the enrichment of complement cascade). We also analysed spatial ecosystem with the distribution of cells in Figure 4, and we added a supplementary Figure for clustering analysis as shown in Figure R13 below.

We added to the Discussion session the following:

“Notably, STImage identifies the most predictable genes, which are functionally relevant to the respective cancer types, demonstrated as relevant enriched pathways for skin cancer, lung cancer, kidney cancer, breast cancer, as well as non-cancer liver disease. For example, SOX10 (a highly sensitive and specific nuclear marker for melanocytic lesions, including primary and metastatic melanoma, PRAME (overexpressed in many melanomas but not in benign nevi or normal melanocytes, a marker and a target for immunotherapy and vaccine-based approaches in melanoma treatment), and S100B (a sensitive marker for melanocytic differentiation in tissue samples) were the top genes with the highest prediction accuracy for melanoma. Similarly, for breast cancer, the top predictable genes include ESR1 (major driver in hormone receptor positive breast cancer), HER2 (ERBB2, a member of the EGFR family of receptor tyrosine kinases, associated with higher cancer progression and recurrence), GATA3 (strong correlation with ER-positive tumours and favourable prognosis), and immune markers like CD45 (marking immune infiltration). We assessed potential clinical utilities through survival analyses and patient stratification for drug responses. The predicted results based on the images were able to classify long vs short survival and complete vs partial responses.”

To further evaluate the biological applicability of the gene expression prediction model on HE images, we performed clustering on the predicted gene expression data to analyse tissue heterogeneity. Overall, the clustering results identify the structural organisation of skin layers (Figure R13A) with expected gene markers. Notably, Cluster 1 corresponds to the melanoma region. We further utilised a set of marker genes associated with epidermal genes, dermal fibroblast genes, and sweat gland genes to characterise clusters. We found that epidermal gene markers were enriched in Cluster 0, while sweat gland genes were enriched in Cluster 5 (Figure R13B). The spatial patterns of the clusters generally align with the HE morphology.

Figure R13. Assessing the prediction of biologically relevant markers and tissue regions. a) Prediction of the three markers for melanoma, PRAME, SOX10 and S100B. Two samples are shown. For each sample, the ground truth values are from spatial transcriptomics, while the predicted are the results from STImage prediction. b) and c) show clustering analysis using predicted values from STImage. Clustering results and gene marker show epidermal layers (cluster 0) and dermal layers with melanocyte/melanoma (1, 2, 9, 12) and sweat gland (clusters 4, 5).

To further assess the clinical utilities, we added to our previous survival analysis an independent assessment of the ability to predict responses to drugs. Here we analysed a new cohort of breast cancer patients with treatment response information. We found that, using our predicted values of the 1139 functional genes based on H&E images, we could stratify patient responses with $AUC = 0.70$, compared with using the bulk RNA-seq with $AUC = 0.78$ (Figure R14).

Figure R14. Assessing the ability of STimage prediction to stratify patient responses to drug treatment. a) The patient cohort with response status on the left [responders (pCR.RD), partial responders (pCR) and not applicable (NA)] and tumour stage on the right (from stage T1 to T4). b) Assessing the regression model using PCC. Box plot showing gene-wise PCC per sample, computed between predicted gene expression from H&E images and matched bulk gene expression across 160 samples. c) Each dot represents a patient. AUC comparison of true and predicted gene expression to classify patient response to drugs using SVM.

Reviewer #3 (Remarks on code availability):

On github, but could be more clearly organized
 We have organised the code.

Reviewer #4 (Remarks to the Author): Expert in cancer digital pathology, AI and machine learning; replaces Reviewer #2

All comments of the reviewer 2 has been addressed.

We thank the reviewer 4 for confirming that our work has met the requirements from the perspectives of digital pathology and AI/ML.

Reference

- Aran, Dvir, Marina Sirota, and Atul J. Butte. 2015. "Systematic Pan-Cancer Analysis of Tumour Purity." *Nature Communications* 6 (December):8971.
- Hörst, Fabian, Moritz Rempe, Lukas Heine, Constantin Seibold, Julius Keyl, Giulia Baldini, Selma Ugurel, et al. 2024. "CellViT: Vision Transformers for Precise Cell Segmentation and Classification." *Medical Image Analysis* 94 (May):103143.
- Yoshihara, Kosuke, Maria Shahmoradgoli, Emmanuel Martínez, Rahulsimham Vegesna, Hoon Kim, Wandaliz Torres-Garcia, Victor Treviño, et al. 2013. "Inferring Tumour Purity and Stromal and Immune Cell Admixture from Expression Data." *Nature Communications* 4:2612.

REVIEWER COMMENTS

Reviewer #1 (Remarks to the Author):

The authors have improved the study, the reviewer still has following comments based on their revised manuscript.

1. The authors suggested the relatively low Pearson correlation coefficients (PCC) observed in several benchmark datasets due to insufficient training sample size. However, they proceed to perform downstream analyses using these potentially inaccurate predictions. As the task of inferring gene expression levels from H&E images is inherently a regression problem, the use of Intersection over Union (IoU) — which binarizes continuous gene expression values — may not appropriately reflect the accuracy of regression outputs. It is strongly recommended that the authors report additional regression metrics such as Mean Absolute Error (MAE) or Mean Squared Error (MSE) to provide a more comprehensive and quantitative assessment of prediction accuracy.

We appreciate the reviewer's suggestion for adding more regression metrics for each dataset. Previous works such as STNet and Hist2ST use MAE or MSE as loss functions to optimise model parameters. Thus, these cannot be used as performance metrics for benchmarking. In line with the reviewer's recommendation, we will report MAE and MSE for all datasets as additional regression metrics (see **Figure R1** and the accompanying supplementary CSV file). These metrics evaluate point-wise accuracy within each sample but do not capture gene-wise trends across spots within a sample. The PCC provides a reasonable measure for this aspect and is widely used in previous works. However, PCC, MAE and MSE do not account for spatial context, which is central to our use case. We therefore further assess the IoU, which quantifies the overlap of high-expression domains. One of the reasons we use IoU is that this metric is intuitively more similar to tissue assessment by pathologists, who classify tissue regions in a manner akin to IoU rather than at a single tissue location. MWe also report Moran's I (see Figure S2), which measures concordance in spatial autocorrelation between predicted and observed gene expression.

Figure R1. Comparison of regression performance metrics (MAE, MSE, PCC) across multiple spatial transcriptomics datasets. Boxplots show mean absolute error (MAE), mean squared error (MSE), and Pearson correlation coefficient (PCC) for predicted versus observed gene expression values across samples within each dataset. Panels correspond to (top row) HER2ST breast cancer samples (Legacy ST), (second row) skin melanoma samples (Visium), (third row) kidney samples (Visium), and (bottom row) liver samples (Visium).

2. This study predominantly highlights genes with strong prediction performance, which appear to coincide with spatial patterns of H&E staining intensity. This raises the concern of potential model bias, specifically, whether STImage disproportionately favors genes with highly localized expression patterns that align with histological contrast. It would be important for the authors to clarify whether STImage struggles with genes exhibiting broadly distributed expression patterns. Furthermore, is there a correspondence between high Pearson correlation coefficients and high spatial overlap of predicted versus true gene expression domains? For genes with low PCC, is inaccuracy primarily due to incorrect prediction of spatial distribution or due to inaccurate expression value estimation?

We appreciate the reviewer's concern about potential bias towards genes with high histological contrast. We evaluated STImage on three gene sets chosen to be independent of H&E intensity patterns: (i) functional genes drawn from immuno-oncology and cell-type marker panels that were selected for biological relevance rather than morphology; (ii) the top 1,000 highly variable genes defined by expression variance in the matrix, not by spatial variability; and (iii) a clinically used 14-gene panel.

To address the concern regarding potential bias towards spatial patterns of H&E staining intensity, we first examined the genes with high prediction accuracy and low correlation with H&E image intensity (Figure R2). The results show that the model is not learning H&E intensity-related features for those highly predictable genes.

Predicted CCL19 expression

Ground truth CCL19 expression

Predicted HLA-DPA1 expression

Ground truth HLA-DPA1 expression

Predicted JCHAIN expression

Ground truth JCHAIN expression

Predicted LYZ expression

Ground truth LYZ expression

Figure R2. examining relationship between gene expression predictions and H&E image intensity, along with spatial expression maps for genes in the HER2ST dataset. For each gene with high PCC and low correlation with H&E image intensity (CCL19, HLA-DPA1, JCHAIN, LYZ), the left panels show scatter plots of predicted expression versus median RGB intensity per tile, annotated with evaluation metrics (IoU, MAE, MSE and PCC). The middle and right panels show spatial expression plots for predicted and ground truth expression, respectively. Warmer colours indicate higher expression levels.

We further performed a more systematic and comprehensive analysis of the relationship between prediction accuracy and the correlation of both true and predicted gene expression with median H&E intensity across four datasets (her2st, skin, kidney, liver), evaluating four metrics (MAE, MSE, PCC, IoU), as shown in Figure R3. We observed dataset-specific trends. In the HER2ST and liver datasets, prediction accuracy tends to be higher for genes with moderate to high correlation with H&E intensity. This could indicate that H&E intensity dominates the histomorphological features. Conversely, for the skin and kidney datasets, prediction performance appears largely independent of H&E intensity correlation, indicating that STImage is not systematically biased towards HE intensity-linked genes. For genes with low PCC, this can be due to both inaccurate spatial distribution and inaccurate expression-value estimation. We also observe the key marker gene *KRT5* for myoepithelial cells in Figure 1C. The predicted pattern matches the ground truth, is expressed in the basal cell layer of DCIS, and shows no correlation with H&E image intensity. It is well recognised that not all genes show distinct spatial structures or a strong relationship to histological morphology, which inherently constrains their predictability.

Figure R3. Relationship between prediction accuracy, gene expression, and histology image intensity across datasets for top 100 most predictable genes. Each scatter plot shows the correlation of predicted gene expression with the median H&E intensity (y-axis) versus the correlation of true gene expression with the median H&E intensity (x-axis) for each top predictable genes in four datasets (HER2ST, skin, kidney, and liver). Point colours indicate different regression metrics(MAE, MSE, PCC and IoU).

3. Multiple previous models incorporate spatial relationships among tissue tiles either through architectural design or as part of the input features, yet STImage appears to omit such explicit spatial encoding. Despite this, the model achieves better predictive performance across multiple benchmarks. It

would be valuable for the authors to discuss the implications of this design choice. Does the model implicitly capture spatial dependencies through other means, or is such information not critical for the tasks addressed?

We thank the reviewer for this thoughtful comment. We agree that spatial relationships are important. In the gene expression prediction task, gene expression values are spot-level aggregates at relatively coarse scales (for example, $\sim 55 \mu\text{m}$ in Visium and $\sim 100 \mu\text{m}$ in legacy ST). Much of the morphologic context and local cell organisation relevant to these gene expression values are already contained within each spot-level image tile. So explicit cross-spots graphs can mix signals over large distances that may not correspond to the target spot and can introduce noise unless very large training sets are available. STimage therefore prioritises robust estimation by modelling gene counts with a negative binomial likelihood rather than using a fixed point estimate, and by employing deep ensembles, which we demonstrate improves performance and stability across benchmarks.

Although we do not impose an explicit spot-level graph to capture the nearby spatial information, in the updated FSTimage variant, we replace image features with foundation-model embeddings (ViT based) trained self-supervised on millions of whole slide images; these embeddings are known to encode multi-scale contextual structure and further improve robustness.

4. This study lacks a direct comparison between STimage's survival prediction performance and existing H&E-based models, such as IGIDL survival model. The authors primarily compare their own univariate and multivariate models, which limits the reader's ability to contextualize the improvements achieved. Furthermore, the rationale for performing subtype-specific survival modeling in breast cancer cohorts is not clearly articulated. Would a pan-subtype survival model (without stratifying by subtype) result in inferior performance, and if so, could this be quantified and discussed?

We thank the reviewer for this constructive comment. The IGI-DL survival model implements a graph-based deep learning approach that takes as input both IGI-DL-predicted gene expression and patient-level clinical variables (e.g., age, race, gender, AJCC pathological stage). The scope of this model is therefore broader than ours, which intentionally evaluates survival prediction performance based solely on predicted gene expression to isolate and quantify the prognostic value of gene expression features derived from histopathology.

Moreover, we attempted to reproduce the IGI-DL survival model following the authors' GitHub repository. While we were able to perform the initial steps (whole slide image pre-processing, patch tiling, stain colour normalisation, and nuclei segmentation), we encountered reproducibility issues at the graph construction stage. Specifically, the pipeline is built for Visium spatial transcriptomics data and relies on Visium spot coordinates, log transform gene expression matrix and hard-coded parameters (e.g., the neighbour filtering distance threshold is $20 \mu\text{m}$, corresponding to $0.5 \mu\text{m}$ per pixel for graph construction). The Visium pipeline uses $55 \mu\text{m}$ tiling (matching the Visium spot size), whereas TCGA uses $100 \mu\text{m}$ tiling). These constraints prevented us from benchmarking with the IGI-DL model on the TCGA dataset.

Instead, we benchmarked four models, representing the three most widely used architectures: CNN-based (ST-Net), ViT-based (HisToGene, Hist2ST), and capsule-net/graph-based (THIToGene). To ensure fair benchmarking, we used four-fold cross-validation for each method to select the best model trained on the Her2ST spatial transcriptomics dataset for the top 1,000 HVGs (737 remaining after quality control, i.e., genes expressed in fewer than 1,000 spots were removed). The model was then used to make predictions on 1,253 TCGA samples (all of TCGA-BRCA with survival information and bulk RNAseq data). Among the four comparisons, including: 1) all samples, 2) HER2-positive, 3) Luminal, and 4) TNBC, the STImage outperforms the all the other four models in 1, 2, and 3, while it ranked the second in the 4 comparison (**Figure R4**). Notably, in all cases, STImage can still stratify patients into high- and low-risk groups across (**Figure R4**).

Regarding the reviewer's question on breast cancer subtype-specific modelling, we stratified the survival models by molecular subtypes, which reflect the well-established prognostic categories to display information that is more clinically relevant for each subtype (ER-positive, HER2-positive and triple-negative). A pan-subtype model would aggregate biologically distinct cohorts, potentially diluting subtype-specific prognostic signals. To address the reviewer's request, we have added a pan-subtype model for comparison and we still observed significant stratification between high- and low-risk groups for the combined category (**Figure R4**). **We have updated the main Figure X accordingly.**

Fig R4. Benchmarking of survival analysis. a) Kaplan-Meier survival curves for true and predicted gene expression from different models for all samples and three subtypes of breast cancer using the predicted gene expression from STImage. b) C-index comparison between different models for all samples and three subtypes of breast cancer.

We have also updated the figure 5 in the manuscript, to have a pan-subtype survival model.

Reviewer #3 (Remarks to the Author):

I appreciate the thorough responses provided by the authors addressing my concerns regarding the biological applications and user accessibility of the STImage tool suite.

Overall, I am satisfied with the responses and the revisions made by the authors.

Reviewer #3 (Remarks on code availability):

The code is easier to navigate on github

We are glad that our previous response satisfied the reviewer, and we thank the reviewer for their effort in testing our GitHub repository.